# Deciphering the CircRNA-Regulated Response of Western Honey Bee (*Apis mellifera*) Workers to Microsporidian Invasion

**DOI:** 10.3390/biology11091285

**Published:** 2022-08-29

**Authors:** Huazhi Chen, Xiaoxue Fan, Wende Zhang, Yaping Ye, Zongbing Cai, Kaiyao Zhang, Kuihao Zhang, Zhongmin Fu, Dafu Chen, Rui Guo

**Affiliations:** 1College of Bee Science, Fujian Agriculture and Forestry University, Fuzhou 35002, China; 2Apitherapy Research Institute, Fujian Agriculture and Forestry University, Fuzhou 35002, China

**Keywords:** western honey bee, circular RNA, *Vairimorpha ceranae*, noncoding RNA, host–pathogen interaction, immune response

## Abstract

**Simple Summary:**

The western honey bee (*Apis mellifera*) is commercially kept for crop pollination and production of bee products in China and many other countries. Currently, the circRNA-regulated response of *A. mellifera* to *Vairimorpha ceranae* infection is completely unknown. Here, characterization of circRNAs engaged in the response of *A. mellifera* workers to *V. ceranae* infection was conducted for the first time, followed by an in-depth investigation of the regulatory role of the hosts’ differentially expressed circRNAs (DEcircRNAs). Our findings not only revealed the underlying molecular mechanism of host response, but also offered novel insights into honey bee–microsporidian interaction.

**Abstract:**

*Vairimorpha ceranae* is a widespread fungal parasite of adult honey bees that leads to a serious disease called nosemosis. Circular RNAs (circRNAs) are newly discovered non-coding RNAs (ncRNAs) that regulate biological processes such as immune defense and development. Here, 8199 and 8711 circRNAs were predicted from the midguts of *Apis mellifera ligustica* workers at 7 d (Am7T) and 10 d (Am10T) after inoculation (dpi) with *V. ceranae* spores. In combination with transcriptome data from corresponding uninoculated midguts (Am7CK and Am10CK), 4464 circRNAs were found to be shared by these four groups. Additionally, 16 circRNAs were highly conserved among *A. m. ligustica*, *Apis cerana cerana*, and *Homo sapiens*. In the Am7CK vs. Am7T (Am10CK vs. Am10T) comparison group, 168 (306) differentially expressed circRNAs (DEcircRNAs) were identified. RT-qPCR results showed that the expression trend of eight DEcircRNAs was consistent with that in the transcriptome datasets. The source genes of DEcircRNAs in Am7CK vs. Am7T (Am10CK vs. Am10T) were engaged in 27 (35) GO functional terms, including 1 (1) immunity-associated terms. Moreover, the aforementioned source genes were involved in three cellular immune-related pathways. Moreover, 86 (178) DEcircRNAs in workers’ midguts at 7 (10) dpi could interact with 75 (103) miRNAs, further targeting 215 (305) mRNAs. These targets were associated with cellular renewal, cellular structure, carbohydrate and energy metabolism, and cellular and humoral immunity. Findings in the present study unraveled the mechanism underlying circRNA-mediated immune responses of western honey bee workers to *V. ceranae* invasion, but also provided new insights into host–microsporidian interaction during nosemosis.

## 1. Introduction

*Apis mellifera ligustica*, a well-known subspecies of *Apis mellifera*, is commercially kept for crop pollination and production of bee products in China and other countries [1]. As a eusocial insect, the honey bee is susceptible to infection by an array of pathogens such as bacteria, fungi, and viruses. Among them, *Vairimorpha ceranae* is an obligate intracellular fungal parasite that mainly infects the midgut tissue of adult honey bees and causes a serious chronic disease called nosemosis, which poses a huge threat to the beekeeping industry [2]. *V. ceranae* is among the most common parasites in *A. mellifera*, with a worldwide distribution [3]. *V. ceranae* infection causes lifespan reduction, energy stress, immunosuppression, apoptosis inhibition, and imbalance of midgut epithelial cell renewal in western honey bees [4,5,6,7,8]. It can also jeopardize bee hosts together with other biotic or abiotic factors, severely damaging bee health and beekeeping production [9]. Evidently, *V. ceranae* has proved to be relevant to colony collapse disorder (CCD) [10,11], which severely influences many aspects of the colony, including reproductive rate, survival rate, absconding rate, etc. [12,13].

In recent years, several omics studies have been conducted to survey the *V. ceranae*-response of *A. mellifera* based on next-generation sequencing and bioinformatics [9,14,15,16,17,18]. For example, Chaimanee et al. analyzed the expression profiling of immune genes in adult workers of *A. mellifera* at 3, 6, and 12 d after infection (dpi) with *V. ceranae* and observed that humoral immune-associated genes such as defensin, abaecin, apidaecin, and hymenoptaecin were significantly down-regulated at 3 dpi and 6 dpi, whereas the expression level of vitellogenin was not changed, suggestive of host immunosuppression due to *V. ceranae* infection [14]. More recently, our group conducted a transcriptomic investigation of the immune responses of *A. m. ligustica* workers to *V. ceranae* invasion and revealed host cellular and humoral immune responses by analyzing the differential expression patterns of genes and miRNAs in the midguts of *A. m. ligustica* workers challenged with *V. ceranae* [18,19].

As a new member of the noncoding RNA (ncRNA) world, circular RNAs (circRNAs) were initially thought to be byproducts of gene transcription without biological function [20]. CircRNA has the characteristics of species conservation, stability, and spatiotemporal expression specificity [21]. Thanks to the rapid development and application of high-throughput sequencing technology and corresponding bioinformatics, an increasing number of circRNAs are being identified in various animal, plant, and microorganism species, such as *Homo sapiens* [22,23], *Mus musculus* [22], *Triticum aestivum* L [24], *Oryza sativa* [25], *Drosophila melanogaster* [26], *Bombyx mori* [27], *A. mellifera* [28], *Apis cerana* [29], *V. ceranae* [30], and *Ascosphaera apis* [31]. Accumulating experimental and computational evidence have indicated the participation of circRNAs in various biological processes [21] through regulating the transcription of source genes [32,33], acting as a “sponge” for microRNA (miRNA) [34] and competing endogenous RNA (ceRNA) [35] or interacting with various proteins [35]. Owing to their covalently closed structure, circRNAs can resist digestion by RNase R and hence are ideal biomarkers and therapeutic targets for disease diagnosis and treatment [36]. More recently, several studies suggested that circRNAs containing internal ribosome entry sites (IRES) or N6-methyladenosine (m6A) methylation sites could synthesize biologically active peptides or proteins [37,38].

In honey bees, only three circRNA-associated studies have been reported [28,29,39]. In a previous work, we conducted deep sequencing of the midguts of *A. m. ligustica* workers at 7 dpi and 10 dpi with *V. ceranae* and corresponding uninoculated midguts using strand-specific cDNA library-based RNA-seq and identified 10,833 novel circRNAs based on transcriptome data from control groups; it was found that these circRNAs were 15–1000 nt in length and the main type of circularization was annotated exon circRNA [40]. In human, chicken, and silkworm, it has been documented that circRNAs are likely to regulate the host response to pathogenic microorganisms [41,42,43,44]. However, there is no report regarding circRNAs engaged in honey bee–parasite/pathogen interaction until now, and little is known about the regulatory role of circRNAs in the western honey bee responding to *V. ceranae* invasion.

Our group previously performed whole transcriptome sequencing of *V. ceranae*-inoculated midguts of *A. m. ligustica* workers at 7 dpi and 10 dpi and corresponding uninoculated midguts using a combination of strand-specific cDNA library-based RNA-seq and small RNA-seq (sRNA-seq); conducted transcriptome-wide identification of host mRNAs, miRNAs, and long noncoding RNAs (lncRNAs); and performed a systematic investigation of the differential expression profiles of genes and ncRNAs in host midguts’ responses to fungal infection [18,19,45]. Here, based on the obtained high-quality whole transcriptome datasets, circRNAs in the *V. ceranae*-inoculated midguts of *A. m. ligustica* workers were predicted using bioinformatics followed by sequence conservation analysis. Additionally, differentially expressed circRNAs (DEcircRNAs) were identified and their source genes were then annotated. Moreover, circRNA-miRNA and circRNA-miRNA-mRNA networks were investigated, and circRNAs associated with host cellular and humoral immune were further explored. To our knowledge, this is the first documentation of the circRNA-mediated immune response of the honey bee to *V. ceranae* infection, which offers not only a key foundation for clarifying the mechanism underlying host immune response, but also a new insight into honey bee–microsporidian interaction.

## 2. Materials and Methods

### 2.1. Biological Samples

*A. m. ligustica* workers were taken from healthy colonies reared in the teaching apiary (119.2369° E, 26.08279° N) of the College of Animal Sciences (College of Bee Science), Fujian Agriculture and Forestry University. Clean spores of *V. ceranae* were previously prepared using the Percoll discontinuous density gradient method [30,46] and kept in the Honey Bee Protection Laboratory of the College of Animal Sciences (College of Bee Science), Fujian Agriculture and Forestry University.

### 2.2. Source of Whole Transcriptome Data

In our previous study, it was found that the cumulated mortality of *A. m. ligustica* workers in the *V. ceranae*-inoculated group was significantly higher than that of workers in the control group at both 7 dpi and 10 dpi [45]. Hence, *V. ceranae*-inoculated workers’ midgut tissues at 7 dpi and 10 dpi and the corresponding uninoculated midgut tissues were harvested [18,47]. Briefly, (1) newly emerged workers were put into sterilized plastic cages (20 per group) with a feeder containing a 50% (*w*/*v*) sucrose solution inserted into each cage and incubated at 34 ± 0.5 °C for 24 h; (2) after 2 h of starvation, a single worker was immobilized for artificial inoculation and workers in treatment groups were each fed with 5 μL of 50% (*w*/*v*) sucrose solution containing 1 × 10^6^
*V. ceranae* spores [45], whereas workers in the control groups were each fed with 5 μL 50% (*w*/*v*) sucrose solution without spores; (3) after 24 h of inoculation, workers in both treatment and control groups were fed with a feeder containing 1 mL of 50% (*w*/*v*) sucrose solution without spores for 24 h, and the feeder was replaced with a new one every 24 h; each cage was checked daily and any dead bees were removed; (4) the midgut tissues of workers (n = 9) in the treatment groups and control groups at 7 dpi and 10 dpi were harvested, quickly frozen in liquid nitrogen, and stored at −80 °C until deep sequencing. There were three biological replicate cages for each treatment (control) group. Midgut samples collected at 7 dpi in the control group and treatment group were respectively termed as Am7CK (Am7CK1, Am7CK2, and Am7CK3) and Am7T (Am7T1, Am7T2, and Am7T3), while midgut samples collected at 10 dpi were respectively termed as Am10CK (Am10CK1, Am10CK2, and Am10CK3) and Am10T (Am10T1, Am10T2, and Am10T3). The aforementioned 12 midgut samples were previously sequenced using strand-specific cDNA library-based RNA-seq technology [45,48]. RNA isolation, strand-specific cDNA library construction, and deep sequencing were performed following our previously described method [29,45,48]. The raw reads were uploaded to the Short Read Archive (SAR) database of the NCBI under BioProject number PRJNA406998. Quality control of raw reads was previously carried out, and the results suggested that approximately 273.27 Gb of raw reads were generated from RNA-seq and 271.40 Gb of clean reads were obtained after strict quality control; on average, the ratio of clean reads among raw reads in every group was 99.32%; for Q20 and Q30, they were 97.34% and 93.82%, respectively [18]. Thus, the gained high-quality transcriptome data could be used for identification and investigation of circRNAs, target prediction and analysis, and construction and analysis of ceRNA regulatory networks in the present study.

Meanwhile, the aforementioned 12 midgut samples were subjected to RNA extraction, cDNA library construction, and high-throughput sequencing using sRNA-seq technology [19]. Briefly, (1) the total RNA of each sample was extracted with TRIzol Reagent (Invitrogen, Carlsbad, CA, USA) and DNA contaminants were then removed with RNase-free DNase I (TaKaRa, Beijing, China); the purified RNA quantity and quality were checked using a Nanodrop 2000 spectrophotometer (Thermo Fisher, Waltham, MA, USA) and the integrity of the RNA samples was evaluated using an Agilent 2100 bioanalyzer (Agilent Technologies, Santa Clara, CA, USA), and only values of 28S/18S ≥ 0.7 and RIN ≥ 7.0 were considered as qualified for the subsequent small RNA library construction; (2) RNA molecules with a size distribution among 18–30 nt were enriched using agarose gel electrophoresis (AGE) and then ligated with 3′ and 5′ RNA adaptors followed by reverse transcription and enrichment of fragments with adaptors on both ends via PCR; (4) the subsequent cDNAs were purified and enriched using 3.5% AGE to isolate the expected size (140–160 nt) of the fractions and eliminate unincorporated primers, primer dimer products, and dimerized adaptors; (5) the 12 cDNA libraries were sequenced on an Illumina MiSeq^TM^ platform using the single-end technology by Gene Denovo Biotechnology Co. (Guangzhou, China). Raw reads were submitted to the SRA database in NCBI under BioProject number PRJNA408312. Quality control of the raw reads was conducted previously, and the results indicated that approximately 23.18 Gb of raw reads were produced from sRNA-seq and about 17.85 Gb of clean tags were gained after data quality control; on average, the ratio of clean tags among raw reads was 77.04%; for Q20 and Q30, they were 97.34% and 93.82%, respectively [19]. Therefore, the sRNA data with high quality could be used for prediction of circRNA-targeted miRNAs and miRNA-targeted mRNAs and analysis of circRNA-miRNA and circRNA-miRNA-mRNA networks in this current work.

In parallel, following the aforementioned protocol, the midgut tissues of *A. m. ligustica* workers (n = 9) at 7 dpi and 10 dpi with *V. ceranae* spores and corresponding un-inoculated midgut tissues were previously prepared and conserved in a −80 °C ultra-low-temperature freezer (Sanyo, Osaka, Japan) and were used for PCR, stem-loop RT-PCR, and real-time quantitative PCR (RT-qPCR) in this current work.

### 2.3. Identification and Conservative Analysis of CircRNAs

CircRNAs were identified according to our previously described protocol [29]. Firstly, the clean reads were mapped to the *A. mellifera* genome (assembly Amel_4.5) [49] with TopHat software (version 2.0.3.12) [50]. Next, 20 nt from the 5′ and 3′ ends of unmapped reads were extracted and aligned independently to reference sequences using Bowtie2 (version 2.2.8) [51]. Ultimately, the unmapped anchor reads were submitted to find_circ software (version 1.1) [22] for the identification of the following circRNA criteria: breakpoints = 1; edit ≤ 2; n uniq > 2; anchor overlap ≤ 2; best qual A > 35 or best qual B > 35; n uniq > int (samples/2); and circRNA length < 100 kb.

To rule out circRNAs from *V. ceranae*, we used a similar circRNA identification protocol to identify *V. ceranae* circRNAs from Am7T and Am10T. Subsequently, we compared the anchor reads of the circular RNA of the two species separately and found no common circRNAs (Appendix A).

To investigate the conservation of circRNAs in *A. m. ligustica* and other species, sequences of the identified circRNAs’ source genes were aligned to those of source genes of the previously identified circRNAs in *Apis cerana cerana* [29] and *H. sapiens* [35,52] with the blastn tool (https://blast.ncbi.nlm.nih.gov/Blast.cgi, accessed on 16 May 2019) using the default parameters; this was performed on 16 May 2019.

### 2.4. Investigation of DEcircRNAs and Their Source Genes

The expression level of each circRNA was calculated using reads per million (RPM) = 10^6^ C/N (where C is the number of back-spliced junction reads that uniquely aligned to a circRNA and N is the total number of back-spliced junction reads). DESeq software (Version 1.28) [53] was employed to identify significant DEcircRNAs in the Am7CK vs. Am7T and Am10CK vs. Am10T comparison groups according to the threshold of |log_2_(Fold change)| = |log_2_FC| ≥ 1 and *p* ≤ 0.05.

CircRNA can regulate source gene expression via interaction with RNA polymerase II, U1 micronucleoprotein, or a gene promoter [32,33]. In this study, source genes of DEcircRNAs were predicted by aligning anchor reads at both ends to the *A. mellifera* genome (assembly Amel_4.5) [49] using the Bowtie tool [51]. Gene Ontology (GO) term analysis for source genes was performed with the DAVID gene-annotation tool (Frederick, MD, USA) (http://david.abcc.ncifcrf.gov, accessed on 16 May 2019) [54]. A two-sided Fisher’s exact test was performed to classify the GO category, while the FDR was calculated to correct the *p*-value [55]. GO terms with a *p*-value < 0.05 were considered to be statistically significant. In addition, a pathway analysis was performed by annotating source genes to the Kyoto Encyclopedia of Genes and Genomes (KEGG) database (http://www.genome.jp/kegg, accessed on 16 May 2019) [56]. The significance threshold was defined by a *Q*-value < 0.05.

### 2.5. Target Prediction and Regulatory Network Analysis of DEcircRNAs

The target miRNAs of DEcircRNAs were predicted using TargetFinder software [57]. Following the cutoff of *p* ≤ 0.05 and free energy ≤35, potential target miRNAs were further extracted to construct a DEcircRNA-miRNA regulatory network. The target mRNAs of miRNAs targeted by DEcircRNAs were then predicted with TargetFinder software, followed by construction of a DEcircRNA-miRNA-mRNA regulatory network. The regulatory networks were visualized utilizing Cytoscape software [58]. Additionally, the target mRNAs within the DEcircRNA-miRNA-mRNA network were annotated to the GO and KEGG databases with the Blast tool. Based on the function and pathway annotations and our previous results [18,19,45], the DEcircRNA-DEmiRNA-DEmRNA regulatory network relevant to host cellular and humoral immunity was further investigated and then visualized with Cytoscape.

### 2.6. PCR and Sanger Sequencing Confirmation of Novel circRNAs

In order to verify the expressions of miRNA and circRNA, an additional independent experiment was performed in which workers’ midgut samples were collected for the extraction of total RNA and genomic DNA according to the steps in Section 2.2. Three circRNAs (novel_circ_004065, novel_circ_002199, and novel_circ_005784) were randomly selected from specific circRNAs in the Am7T and Am10T groups for PCR and Sanger sequencing. Theoretically, for circRNA, convergent primers should amplify products from both the cDNA template and the gDNA template, but divergent primers should only amplify products from the cDNA template [29]. Following our previously described method [29,40], specific convergent primers and divergent primers (shown in Appendix A) for the three circRNAs mentioned above were designed using DNAMAN 8 software (Lynnon Biosoft, San Ramon, CA, USA) and synthesized by Shanghai Sangon Biological Co., Ltd., Shanghai, China. The total RNA of the previously prepared workers’ midguts (described in Section 2.2) at 7 dpi and 10 dpi were respectively isolated with an AxyPre RNA extraction kit (Axygen, Hangzhou, China) and divided into two portions; one portion was digested with 3 U/mg RNase R (Geneseed, Guangzhou, China) at 37 °C for 15 min to remove the linear RNA and then synthesize the first-strand cDNA via transcription with random primers, whereas the other portion was subjected to reverse transcription using Oligo (dT) 18 as templates to synthesize the first-strand cDNA. The genomic DNA (gDNA) of workers’ midguts at 7 (10) dpi was extracted using the AxyPre DNA extraction kit (Axygen, Hangzhou, China). An equimolar mixture of cDNA derived from workers’ midguts at 7 dpi and 10 dpi was used as a template for PCR amplification, which was conducted on a T100 thermocycler (Bio-Rad, Hercules, CA, USA) in a 20 μL reaction volume containing 1 μL of template, 10 μL of PCR Mixture (TaKaRa, Kusatsu, Japan), 1 μL of upstream primers (10 μmol/L), 1 μL of downstream primers (10 μmol/L), and 7 μL of ddH_2_O. The PCR conditions were set as follows: pre-denaturation at 94 °C for 5 min followed by 36 cycles of denaturation at 94 °C for 50 s, an appropriate annealing temperature (according to the melting temperature of the primer) for 30 s, extension at 72 °C for 1 min, and final extension at 72 °C for 5 min. The PCR products were detected on 1.5% AGE, cloned into pMD-19-T vector, and subjected to Sanger sequencing by Shanghai Sangon Biological Co., Ltd.

### 2.7. Stem-Loop RT-PCR Verification of miRNAs

Ten miRNAs within the DEcircRNA-miRNA-mRNA regulatory network were randomly selected for stem-loop RT-PCR, including five miRNAs targeted by DEcircRNAs in the Am7CK vs. Am7T comparison group (mir-30-x, mir-451-x, mir-29-y, ame-miR-3720, and mir-21-x) and six miRNAs targeted by DEcircRNAs in the Am10CK vs. Am10T comparison group (mir-21-x, mir-146-x, mir-143-y, mir-101-y, mir-462-x, and mir-7975-y). Specific stem-loop primers, specific forward primers, and universal reverse primers (presented in Appendix A) of these miRNAs were designed using DNAMAN 8 software (Lynnon Biosoft, San Ramon, CA, USA). The total RNA of the previously prepared workers’ midguts (described in Section 2.2) at 7 dpi and 10 dpi were respectively extracted using an Eastep^®^ Super Total RNA extraction kit (Promega, Shanghai, China) followed by synthesis of the corresponding cDNA with stem-loop primers. The PCR reaction was carried out in a 20 μL reaction volume containing 1 μL of cDNA template, 1 μL of upstream primers (10 μmol/L), 1 μL of downstream primers (10 μmol/L), 7 μL ddH_2_O, and 10 μL of PCR Mixture (TaKaRa, Japan). The PCR reaction was conducted on a T100 thermocycler (Bio-Rad, Hercules, CA, USA) using the following parameters: pre-denaturation at 94 °C for 5 min; 36 cycles of denaturation at 94 °C for 50 s, annealing at 56 °C for 30 s, elongation at 72 °C for 1 min, and final elongation at 72 °C for 10 min. The amplified products were examined on 2% AGE.

### 2.8. RT-qPCR Validation of DEcircRNAs

Six DEcircRNAs in the Am7CK vs. Am7T comparison group (novel_circ_000705, novel_circ_001195, novel_circ_011173, novel_circ_006925, novel_circ_012352, and novel_circ_012316) and two DEcircRNAs in the Am10CK vs. Am10T comparison group (novel_circ_007686 and novel_circ_011500) were randomly selected for RT-qPCR verification. Specific divergent primers and convergent primers (shown in Appendix A) for DEcircRNAs were designed with DNAMAN 8 software. The total RNA of the previously prepared workers’ midguts (described in Section 2.2) at 7 dpi and 10 dpi were respectively extracted and divided into two parts. One part was digested with 3 U/mg RNase R (Geneseed, Guangzhou, China) at 37 °C for 15 min followed by reverse transcription with a random primer, and the resulting cDNA was used as template for DEcircRNAs; the other part was reversely transcribed with Oligo (dT)18 and the resulting cDNA was used as template for reference gene (*actin*). The reaction system totaled 20 μL and contained 1 μL of forward primer (10 μmol/L), 1 μL of reverse primer (10 μmol/L), 1 μL of cDNA template, 10 μL of SYBR Green Dye, and 7 μL of DEPC water. The reaction was conducted on a QuantStudio3 real-time PCR instrument (ABI, Los Angeles, CA, USA); each reaction was performed in triplicate following the instructions for the SYBR Green Dye Kit (Vazyme, Nanjing, China). The reaction conditions were set as follows: pre-denaturation at 94 °C for 5 min, denaturation at 94 °C for 50 s, and extension at 60 °C for 30 s, for a total of 45 cycles. The relative expression of DEcircRNAs was calculated using the 2^−^^△△^^Ct^ method [59] and presented as relative expression levels from three biological replicates and three technical parallel replicates followed by visualization with GraphPad Prism 7 software (GraphPad, San Diego, CA, USA).

## 3. Results

### 3.1. Overview of the Transcriptome Data Generated from V. ceranae-Inoculated Midguts of A. m. ligustica Workers

On average, there were 236,067,761 and 326,923,515 anchor reads in the Am7T and Am10T groups, respectively, and 16,349,641 and 18,445,112 anchor reads could be mapped to the *A. mellifera* genome (assembly Amel_4.5) (Table 1). Additionally, the Pearson correlation coefficients between the different biological replicas within both the Am7T and Am10T groups were above 0.96 (Appendix A). Our previous results showed that an average of 180,533,501 and 182,634,149 anchor reads were respectively gained from the Am7CK and Am10CK groups, while 18,794,099 and 18,687,848 anchor reads could be mapped to the *A. mellifera* genome (assembly Amel_4.5) [40]; the Pearson correlation coefficients between the different biological replicas were above 0.95 [47]. These results together suggested that the transcriptome data and sample preparation in our study were reasonable and reliable.

### 3.2. Quantity, Expression, and Conservation of A. m. ligustica circRNAs

Here, 8199 and 8711 circRNAs were identified in the Am7T and Am10T groups, respectively. In our previous work, 6530 and 6289 circRNAs were respectively identified in the Am7CK and Am10CK groups [40]. In total, after removing redundant circRNAs, 14,909 nonredundant circRNAs in *A. m. ligustica* were identified using a combination of datasets from *V. ceranae*-inoculated and uninoculated groups. Additionally, 4464 circRNAs were shared by the above-mentioned four groups, while 1398, 1696, 1019, and 1871 circRNAs were specifically transcribed in each group (Figure 1A). Moreover, distinct PCR products with the expected size were amplified with the divergent primers (Figure 1B,C), confirming the true expression of the identified *A. m. ligustica* circRNAs.

The source genes’ sequences of 14,909 nonredundant *A. m. ligustica* circRNAs were aligned with those of circRNAs in *A. c. cerana* [29] and *H. sapiens* [35,52]; the results demonstrated that only 20 (0.13%) *A. m. ligustica* circRNAs had homology with *H. sapiens* circRNAs, whereas as many as 10,226 (68.59%) *A. m. ligustica* circRNAs were homologous to *A. c. cerana* circRNAs (Figure 2). In addition, there were 16 (0.11%) conservative circRNAs among these three species (Figure 2).

### 3.3. DEcircRNAs Involved in Response of A. m. ligustica Workers to V. ceranae Infection

The overall expression level of circRNAs in the Am7CK group was slightly higher than that in Am7T, while circRNAs in the Am10CK group had a slightly lower overall expression than that in the Am10T group (Figure 3A). In total, 168 DEcircRNAs were screened out in the Am7CK vs. Am7T comparison group, including 61 up-regulated circRNAs and 107 downregulated ones (Figure 3B; see also Appendix A). Among these DEcircRNAs, the top three circRNAs with the highest upregulations were novel_circ_004821 (log_2_FC = 16.15), novel_circ_002611 (log_2_FC = 16.08), and novel_circ_000878 (log_2_FC = 16.04); whereas the most downregulated circRNA was novel_circ_014612 (log_2_FC = −16.85), followed by novel_circ_012545 (log_2_FC = −16.81) and novel_circ_005159 (log_2_FC = −16.72) (Appendix A). In the Am10CK vs. Am10T comparison group, 143 up-regulated circRNAs and 163 down-regulated ones were identified (Figure 3B, see also Appendix A). Among them, novel_circ_002577 (log_2_FC = 16.22) was the most highly expressed one, followed by novel_circ_011847 (log_2_FC = 16.07) and novel_circ_007020 (log_2_FC = 16.83); while the most down-regulated circRNAs were novel_circ_006432 (log_2_FC = −17.38), novel_circ_011107 (log_2_FC = −16.16), and novel_circ_002456 (log_2_FC = −15.86) (Appendix A). In addition, a Venn analysis indicated that 9 up-regulated circRNAs and 10 down-regulated circRNAs were shared by both comparison groups (Figure 3C), whereas the numbers of specifically up-regulated and down-regulated circRNAs were 54 (134) and 97 (155) (Figure 3C).

Moreover, back-splicing sites of eight DEcircRNAs were validated using PCR with divergent primers and Sanger sequencing (Figure 4A). The RT-qPCR results suggested that the expression trend of eight DEcircRNAs was in accordance with that detected in the transcriptome data (Figure 4B). These results further proved the reliability of the identified circRNAs and transcriptome data in the present study (Figure 4B).

### 3.4. Functional Annotation of Source Genes of DEcircRNAs

The GO database annotation indicated that the source genes of the DEcircRNAs in workers’ midguts at 7 dpi with *V. ceranae* were engaged in 27 functional terms relevant to biological process, cellular component, and molecular function. In the biological process category, the largest subcategory was cellular process, followed by single-organism process, metabolic process, localization, and biological regulation; in the cellular component category, the most abundant groups were cell part, cell, membrane, membrane part, and organelle; in the molecular function category, the top five subcategories were binding, catalytic activity, transporter activity, nucleic acid binding transcription factor activity, and molecular transducer activity (Figure 5A–C; see also Appendix A). In workers’ midguts at 10 dpi with *V. ceranae*, the source genes of DEcircRNAs could be annotated to 35 functional terms, including 15 biological-process-associated terms such as cellular process, single-organism process, and metabolic process; 12 cellular component-associated terms such as membrane, membrane part, cell, and cell part; and 8 molecular function-associated terms such as binding, catalytic activity, and nucleotide binding transcription factor activity (Figure 5D–F; see also Appendix A).

Further investigation revealed that 21 (63) and 2 (6) source genes of DEcircRNAs in Am7CK vs. Am7T (Am10CK vs. Am10T) comparison group were engaged in cell-renewal-related terms, including cellular process and cell component organization or biosynthesis (Figure 5; see also Appendix A); additionally, 10 (26), 10 (26), 8 (27), 7 (26), 7 (21), 4 (4), and 1 (1) source genes were involved in cell-structure-associated terms such as cell part, cell, membrane, membrane part, and membrane-enclosed lumen (Figure 5; see also Appendix A); moreover, 4 (6) source genes were annotated to response to stimulus, a functional term relative to host immune defense system (Figure 5; see also Appendix A).

### 3.5. KEGG Pathway Annotation of Source Genes of DEcircRNAs

The KEGG database annotation results demonstrated that source genes of DEcircRNAs in the Am7CK vs. Am7T comparison group were associated with 33 pathways, including the endocytosis, FoxO signaling, phagosome, galactose metabolism, and arginine and proline metabolism pathways (Figure 6A; see also Appendix A); while those in the Am10CK vs. Am10T comparison group were related to 43 pathways, including the endocytosis, Hippo signaling, starch and sucrose metabolism, insulin resistance, and purine metabolism pathways (Figure 6B; see also Appendix A).

Further analysis was performed to explore the pathways relative to cell renewal, carbohydrate metabolism, and energy metabolism, and the results suggested that in both comparison groups mentioned above, two (one) and six (two) source genes of DEcircRNAs were involved in the Hippo and Wnt signaling pathways (Figure 6; see also Appendix A); additionally, two, two, one, and one source genes of DEcircRNAs in the Am7CK vs. Am7T group were annotated to galactose metabolism, starch and sucrose metabolism, and fructose and mannose metabolism, whereas only one source gene was annotated to oxidative phosphorylation, a key energy-metabolism-related pathway in eukaryotes (Figure 6A; see also Appendix A). Comparatively, two and four source genes of DEcircRNAs in the Am10CK vs. Am10T group were engaged in galactose metabolism as well as starch and sucrose metabolism (Figure 6B; see also Appendix A).

In addition, the pathways associated with host cellular and humoral immunity were examined, and the results indicated that four, three, and two source genes of DEcircRNAs in the Am7CK vs. Am7T group were annotated to several cellular immune pathways such as the endocytosis, phagosome, and lysosome pathways (Table 2); while eight, two, one, and one source genes of DEcircRNAs in the Am10CK vs. Am10T comparison group were engaged in endocytosis, phagosome, ubiquitin-mediated proteolysis, and metabolism of xenobiotics by cytochrome P450 (Table 2). Intriguingly, the source gene of DEcircRNA in both comparison groups was found to involve only one humoral immune pathway: the FoxO signaling pathway (Figure 6, Table 2).

### 3.6. DEcircRNA-miRNA Regulatory Network Engaged in Response of A. m. ligustica Workers to V. ceranae Invasion

A total of 82 target miRNAs of 122 DEcircRNAs in workers’ midguts at 7 dpi with *V. ceranae* were predicted, among which novel_circ_011088, novel_circ_013731, and novel_circ_009951 had the most targets, amounting to 48, 37, and 22, respectively (Figure 7A; see also Appendix A); additionally, mir-8503-x could be targeted by as many as 16 DEcircRNAs, while novel-m0007-5p and mir-151-x could be targeted by 14 and 11 DEcircRNAs, respectively (Figure 7A, see al Appendix A). In workers’ midguts at 10 dpi with *V. ceranae*, 106 target miRNAs of 234 DEcircRNAs were identified; novel_circ_011577, novel_circ_002577, and novel_circ_012916 had the most targets, reaching 53, 24, and 24, respectively (Figure 7B; see also Appendix A); additionally, mir-8503-x (24 DEcircRNAs), ame-miR-3747b (22 DEcircRNAs), and ame-miR-981 (21 DEcircRNAs) could be targeted by the most DEcircRNAs (Figure 7B; see also Appendix A). Moreover, stem-loop RT-PCR confirmed the expression of 10 target miRNAs within the DEcircRNA-miRNA regulatory networks (Figure 7C).

### 3.7. CeRNA Regulatory Network Associated with V. ceranae Response by A. m. ligustica Workers

The target mRNAs of DEcircRNA-targeted miRNAs were further predicted, and a ceRNA regulatory network analysis demonstrated that 86 DEcircRNAs in the Am7CK vs. Am7T group could bind to 75 miRNAs, further targeting 215 mRNAs (Appendix A); whereas 178 DEcircRNAs in the Am10CK vs. Am10T group could link to 103 miRNAs, further targeting 305 mRNAs (Appendix A).

Functional annotation indicated that the target mRNAs within the ceRNA regulatory network in workers’ midguts at 7 dpi with *V. ceranae* were relative to 33 functional terms, including cellular process, cellular component organization or biogenesis, cell, membrane, response to stimulus, and cell killing (Figure 8A; see also Appendix A); these targets could also be annotated to 41 pathways, including the Hippo and Wnt signaling pathways, amino sugar and nucleotide sugar metabolism, galactose metabolism, oxidative phosphorylation, and several cellular and humoral immune pathways such as endocytosis, phagosomes, ubiquitin-mediated proteolysis, and the FoxO and MAPK signaling pathways (Figure 8B; see also Appendix A). In the workers’ midguts at 10 dpi with *V. ceranae*, the target mRNAs within the ceRNA regulatory network were associated with 28 functional terms, including cellular process, cellular component organization or biogenesis, cell part, membrane part, and response to stimulus (Figure 8C; see also Appendix A); these targets also could be annotated to 47 pathways, including the Hippo and Wnt signaling pathways, amino sugar and nucleotide sugar metabolism, fructose and mannose metabolism, the pentose phosphate pathway, glycolysis, gluconeogenesis, and several cellular and humoral immune pathways such as endocytosis, lysosome, ubiquitin-mediated proteolysis, and the FoxO and MAPK signaling pathways (Figure 8D; see also Appendix A).

Based on our previous studies on DEmRNAs, DEmiRNAs, and DElncRNAs involved in *A. m. ligustica* workers’ midguts responding to *V. ceranae* infection [18,19,45], DEcircRNA-DEmiRNA-DEmRNA regulatory networks associated with host cellular and humoral immune responses were further analyzed. The results indicated that 16 DEcircRNAs in workers’ midguts at 7 dpi with *V. ceranae* could link to 15 DEmiRNAs and further target 10 DEmRNAs relative to endocytosis, ubiquitin-mediated proteolysis, phagosomes, the FoxO signaling pathway, and the MAPK signaling pathway (Figure 9A; see also Table 3), whereas three DEcircRNAs in workers’ midguts at 10 dpi with *V. ceranae* could bind to 26 DEmiRNAs and further target 10 DEmRNAs relative to the lysosome, ubiquitin-mediated proteolysis, endocytosis, MAPK signaling, and FoxO signaling pathways (Figure 9B; see also Table 3).

## 4. Discussion

### 4.1. Number, Conservation, and Expression Pattern of A. m. ligustica circRNAs during V. ceranae Infection

Currently, there are two mainstream strategies to determine circRNA that utilize high-throughput sequencing: one is to construct a strand-specific cDNA library followed by deep sequencing without RNase R digestion, and the other is to digest total RNA with RNase R to remove linear RNA followed by next-generation sequencing [53]. In this work, the former strategy was employed because the interaction between circRNAs and other RNAs such as miRNAs and lncRNAs could not be investigated based on the linear RNA-removal method, although more circRNAs, especially those with low expression levels, could be detected. Here, 8199 and 8711 circRNAs were identified from the Am7T and Am10T groups, respectively; 4464 circRNAs were shared by Am7T and Am10T, while 1389 and 1696 were specifically expressed in these two groups, indicative of the stress-stage-specific expression of circRNAs, which had been reported in other species [21,60]. Additionally, 4 464 common circRNAs were found in the Am7CK, Am10CK, Am7T, and Am10T groups. These shared circRNAs were speculated to play essential roles in not only normal midguts of *A. m. ligustica* workers, but also midguts in the context of *V. ceranae* infection. When combining the circRNAs discovered in the *V. ceranae*-inoculated and uninoculated groups, a total of 14,909 nonredundant *A. m. ligustica* circRNAs were finally identified, which provided a valuable western honey bee circRNA reservoir for further study in the near future. Moreover, conservative analysis suggested that 68.59% of *A. m. ligustica* circRNAs were homologous to *A. c. cerana* circRNAs, while only 0.13% had homology with human circRNAs. This indicated that a majority of circRNAs were conserved between the aforementioned two sister bee species but the conservation of circRNAs in distant species was much lower, which was consistent with findings in other species such as mouse and wheat [61,62]. Intriguingly, 16 circRNAs (novel_circ_002387, novel_circ_008844, novel_circ_008846, novel_circ_008847, etc.) were highly conserved in the three species mentioned above, which was suggestive of a pivotal function in both honey bees and humans, thus deserving additional investigation. Here, 168 and 306 DEcircRNAs were respectively observed in the Am7CK vs. Am7T and Am10CK vs. Am10T comparison groups, including 9 common up-regulated circRNAs and 10 common down-regulated ones; the numbers of unique up-regulated and down-regulated circRNAs were 54 (134) and 97 (155), respectively. This demonstrated that the expression of some circRNAs was altered in the midguts of *A. m. ligustica* workers due to *V. ceranae* infection, indicating their participation in the hosts’ *V. ceranae* response. The results suggested that these common DEcircRNAs may play roles in the host midgut during microsporidian infection, whereas the specific DEcircRNAs were likely to be involved in the host response to *V. ceranae* invasion at various time points.

### 4.2. DEcircRNAs Were Potentially Engaged in the Host Response to V. ceranae Infection by Regulating the Trasncription of Corresponding Source Genes

*V. ceranae* can not only destroy the structure of epithelial cells in the honey bee midgut [9], but also prolong the survival time of inoculated epithelial cells of western honey bee workers by inhibiting apoptosis, thereby exploiting material and energy from host cells for its proliferation [8,63]. Here, we observed that the source genes of DEcircRNAs in workers’ midguts at 7 dpi with *V. ceranae* were engaged in functional terms related to cell renewal such as cellular process (21 source genes) and cellular component organization or biogenesis (2 source genes); 16 source genes (ncbi_550645, ncbi_410860, ncbi_411018, etc.) were annotated to cell-structure-associated functional terms including cell component, cell, membrane, membrane part, organelle, organelle part, and membrane-enclosed lumen; while in workers’ midguts at 10 dpi with *V. ceranae*, more source genes of DEcircRNAs were involved in cellular process (63 source genes) and cellular component organization or biogenesis (6 source genes); similarly, more source genes (49) were annotated to cell part, cell, membrane, membrane part, organelle, organelle part, and membrane-enclosed lumen. Collectively, these results demonstrated that DEcircRNAs in the midguts of *A. m. ligustica* workers probably regulate the transcription of host genes relevant to the renewal and structure of the midgut epithelial cells to cope with the stress and damage to the midguts caused by *V. ceranae*.

In *Drosophila*, it has been proven that the Hippo signaling pathway can sense damage, deliver unpaired cytokines to stem cells, and conduct gut cell renewal [64]. The regeneration of stem cells in the *Drosophila* gut is mainly controlled by the Wnt signaling pathway [65]. Panek et al. found that the regeneration rate of gut stem cells of *A. mellifera* workers at 7 dpi and 14 dpi with *V. ceranae* decreased significantly due to *V. ceranae* reproduction [8], and further revealed that the Hippo and Wnt signaling pathways may jointly regulate stem cell differentiation in the midgut epithelium [9]. Here, we discovered that two (six) and one (two) source genes of DEcircRNAs in the Am7CK vs. Am7T (Am10CK vs. Am10T) comparison groups were involved in the Hippo and Wnt signaling pathways, indicating that host DEcircRNAs may participate in the regeneration of host midgut epithelial cells by regulating the expression of source genes associated with the Hippo and Wnt signaling pathways and further responding to *V. ceranae* infection.

*V. ceranae* lacks mitochondria and hence highly depends on host-produced ATP, which is mainly derived from the metabolism of various sugars [66]. *V. ceranae* infection can increase the expression of genes related to carbohydrate metabolism in honey bees [17] and enhance host sensitivity of low-concentration sugar water and sugar water intake [5]. In insects, oxidative phosphorylation is the major way to synthesize ATP [67]. We previously reported that DElncRNAs and DEmiRNAs in the midguts of *A. m. ligustica* workers may regulate the expression levels of corresponding target genes, further regulating sugar-metabolism-associated pathways such as galactose metabolism and oxidative phosphorylation to respond to energy stress caused by *V. ceranae* invasion [18,19,45]. Here, two source genes of DEcircRNAs (novel_circ_013719, novel_circ_008338, novel_circ_008345, etc.) in workers’ midguts at 7 dpi were engaged in galactose metabolism, including the aldose-reductase coding gene (ncbi_412163) and alpha-glucosidase coding gene (ncbi_411257); additionally, two source genes (the alpha-glucosidase coding gene (ncbi_411257) and the trehalase coding gene (ncbi_410484)) were involved in starch and sucrose metabolism; one source gene (the aldose-reductase coding gene (ncbi_412163)) of novel_circ_013719 was annotated to fructose and mannose metabolism. Comparatively, in workers’ midguts at 10 dpi with *V. ceranae*, two source genes of DEcircRNAs were involved in galactose metabolism, including the alpha-glucosidase-2 coding gene (ncbi_409889) and the alpha-glucosidase coding gene (ncbi_411257); four source genes were annotated to this sugar-metabolism-related pathway, including the UDP-glucuronosyltransferase 2 A3 coding gene (ncbi_409203) and the alpha-glucosidase-2 coding gene (ncbi_409889). Intriguingly, only one source gene (the V-type proton ATPase 116 kDa subunit a coding gene (ncbi_412810)) of novel_circ_006925 in the hosts’ midguts at 7 dpi was annotated to the oxidative phosphorylation pathway, a key energy metabolism pathway in honey bees. In summary, these results indicated that western honey bee workers may regulate sugar metabolism and oxidative phosphorylation by controlling the transcription of corresponding source genes through alteration of the expression of DEcircRNAs, thus participating in the host response to energy stress triggered by *V. ceranae*. However, more efforts are required to clarify the underlying mechanism.

Similar to other insects, honey bees are able to resist pathogen invasion through cellular and humoral immunity at the individual level [68]. For honey bees, the cellular immune system is mainly composed of endocytosis, encapsulation, phagosome, melanization, and enzymatic hydrolysis, whereas the humoral immune system consists of synthesis and secretion of antibacterial peptides [68]. The endocytosis and phagosome pathways are two major cellular immune pathways of honey bees [69]. The lysosome pathway has antifungal and antiviral activities, allowing hosts to dissolve pathogenic proteins delivered by endocytosis and phagosomes [70,71]. As a significant way to clear damaged or unwanted cells, the ubiquitin–proteasome degradation system plays a primary role in cellular processes such as the cell cycle, division, differentiation, development, and immunity [72]. Insect cytochrome P450 can participate in the insect immune response by catalyzing the biosynthesis and degradation of endogenous substances, including ecdysone and juvenile [73]. In the workers’ midguts at 7 dpi with *V. ceranae*, we observed four endocytosis-associated source genes such as the EH domain-containing protein 3 coding gene (ncbi_413012), three phagosome-related source genes including the V-type proton ATPase 116 kDa subunit a coding gene (ncbi_412810), and two lysosome-associated source genes such as the battenin coding gene (ncbi_410860). Comparatively, more source genes (eight) in the hosts’ midguts at 10 dpi were found to enrich the endocytosis pathway; additionally, eight (G-protein-coupled receptor kinase 1 coding gene, etc.), two (ras-related protein Rab-5C coding gene, etc.), one (E3 ubiquitin-protein ligase TRIP12 coding gene), and one (UDP-glucuronosyltransferase-2A3 coding gene) source genes were engaged in the pathways for endocytosis, phagosomes, ubiquitin-mediated proteolysis, and metabolism of xenobiotics by cytochrome P450, respectively; however, no source gene was detected to be involved in the lysosome pathway. *A. c. cerana* is a subspecies of the eastern honey bee (*Apis cerana*) that is widely reared in China and other Asian countries. In another work, we discovered that the source genes of DEcircRNAs in *A. c. cerana* workers’ midguts at 7 dpi with *V. ceranae* were annotated to two immune pathways (for endocytosis and ubiquitin-mediated proteolysis); however, no source genes were enriched in other cellular immune pathways and humoral immune pathways [74]. Together, these results suggested that various host cellular immune pathways were regulated by corresponding DEcircRNAs through modulation of the transcription of source genes during *V. ceranae* infestation.

In the FoxO signaling pathway in humans, FoxO proteins regulate the expression of a series of target genes and participate in the immune response [75]. The FoxO transcription factor in the nucleus promotes transcription of the *Bim* gene, further leading to apoptosis [76]. Here, only one humoral immune-related pathway (the FoxO signaling pathway) was found to be enriched by source genes of DEcircRNAs in the workers’ midguts at both 7 dpi and 10 dpi with *V. ceranae*. Interestingly, no source gene was enriched in the four classic humoral immune pathways of the western honey bee (the Toll, Imd, JAK/STAT, and JNK signaling pathways [77]), which indicated that the FoxO signaling pathway may be employed by western honey bee workers to tackle *V. ceranae* invasion via regulation of the transcription of source genes by corresponding DEcircRNAs. Taken together, these results showed the participation of DEcircRNAs in host cellular and humoral immune responses to *V. ceranae* infection by modulating the transcription of the corresponding source genes. In addition, we previously discovered that target genes related to the cellular and humoral immune pathways (endocytosis, FoxO signaling pathway, etc.) were also regulated by miRNAs and lncRNAs in *A. m. ligustica* workers’ midguts [19,45]. We speculated that western honey bee workers can adopt different strategies to regulate cellular and humoral immune pathways to respond to the same fungal infection.

### 4.3. DEcircRNAs Were Potentially Involved in the Regulation of Host Immune Defense against V. ceranae Infection via ceRNA Network

Accumulating evidence has shown that circRNAs can act as “molecular sponges” to absorb miRNAs, thereby affecting the expression of downstream target genes [78,79]. Huang et al. discovered that circHIPK2 in mice could inhibit miR124-2HG activity by competitively targeting mir124-2HG, resulting in upregulation of the downstream sigma nonopioid intracellular receptor 1 (*SIGMAR1*) gene; knockout of circHIPK2 regulated autophagy and endoplasmic reticulum stress by altering the expression of miR124-2HG and *SIGMAR1*, further suppressing the activation of astrocytes [78]. Li and colleagues found a significant upregulation of CIRS-7 in the esophageal squamous cell carcinoma tissues—overexpression of CIRS-7 in vitro inhibited miR-7-mediated cell proliferation, migration, and invasion, and also inhibited tumor growth and lung metabolism. The authors further revealed that CIRS-7, as a “molecular sponge” of miR-7, played a biological role by reactivating the downstream HOXB13 gene and its NF-*κ*B/P65 pathway [79]. In the past decade, researchers have made an array of advances in the honey bee miRNA field, indicative of the miRNA-mediated regulation in bee neural development [80], labor division [81], caste differentiation [82], and immune defense [19]. It was previously verified that miR-1 is engaged in the immune process of *Drosophila melanogaster* [83] and *Aedes aegypti* [84] in response to infection by various pathogens. In this study, we noted that 10 DEcircRNAs (novel_circ_002002, novel_circ_003045, novel_circ_003465, etc.) in the Am10CK vs. Am10T comparison group could be targeted by miR-1-z, suggesting these DEcircRNAs were likely to exert an miR-1-z-mediated function in host energy metabolism and immune response. Hua et al. (2017) previously found that miR-25 was correlated with the invasion and proliferation of esophageal squamous cell carcinoma (ESCC) cells; the high expression of miR-25 resulted in a decreased survival rate and the alteration of miR-25 significantly suppressed the invasion and proliferation of ESCC cells [85]. Wang’s group revealed that the expression level of miR-92a in HCC cells was significantly higher than that of normal HCC cells; overexpression of miR-92a promoted the growth and invasion of HCC cells, while knockdown of miR-92a played an inhibitory role [86]. In this current work, five and seven DEcircRNAs in the Am7CK vs. Am7T comparison group were found to respectively target miR-25-x and miR-92-x, two miRNAs highly homologous to miR-25 and miR-92a. Xie et al. reported that miR-30a was down-regulated in colorectal cancer (CRC) tissues and cell lines compared with normal rectal tissues and cells, while overexpression of miR-30a in vitro inhibited proliferation of CRC cells and promoted apoptosis of cancer cells; additionally, they validated that miR-30a can bind to the cancer-related ecto-5′-nucleotidase (*CD73*) gene and further revealed that overexpression of *CD73* could rescue miR-30a-induced inhibition of CRC cell proliferation [87]. Here, eight and seven DEcircRNAs in the hosts’ midguts at 7 dpi with *V. ceranae* invasion were observed to respectively target miR-30-x and miR-30-y, two miRNAs that share a high homology with miR-30a; whereas 12 DEcircRNAs in the hosts’ midguts at 10 dpi were jointly linked to miR-30-y. To sum up, these results demonstrated that the corresponding DEcircRNAs of honey bees may be engaged in the host response to *V. ceranae* infection by acting as “molecular sponges” and absorbing miR-25-x, miR-30-y, and miR-92-x. Additional research is needed to disclose the biological function of the aforementioned DEcircRNAs.

Peptidoglycan recognition proteins (PGRPs) can not only act as recognition proteins, but also can degrade pathogenic cell walls via amidase activity [88,89,90]. The potential vital role of PGRPs in recognizing *V. ceranae* infection in the western honey bee was suggested based on transcriptome investigation [17]. In the present study, in the hosts’ midguts at 7 dpi, novel_circ_012410 (log_2_FC = −14.37) was found to potentially regulate the expression of the PGRP-S2 gene (NM_001163716.1) to participate in identifying *V. ceranae* infection. In addition, we found 15 circRNAs (novel_circ_001614, novel_circ_002958, novel_circ_010263, et al.) in the workers’ midguts at 7 dpi that may act as ceRNAs to regulate the expression of a PGRP-LC-encoding gene (XM_006565506.2). The results showed that these DEcircRNAs acting as ceRNAs may be involved in the recognition of *V. ceranae* infection.

During the microsporidian infection process, programmed cell death (PCD) of the honey bee was previously shown to be inhibited by *V. ceranae* to facilitate its own proliferation [7]. Here, in the hosts’ midguts at 7 dpi, novel_circ_014246 (log_2_FC = 2.33, *p* = 0.033736282) was observed to potentially regulate the cell division cycle and expression of the apoptosis regulator protein 1 gene. As ceRNAs, a total of 20 DEcircRNAs were detected to potentially regulate the expression of a gene (XM_016914485.1) encoding apoptosis stimulation of the p53 protein 2-like isoform X3, whereas 15 DEcircRNAs may have regulated the expression of a gene (XM_016912809.1) encoding the apoptotic chromatin condensation inducer in the nucleus-like isoform X1. This indicated that these DEcircRNAs may participate in activating apoptosis in western honey bee workers to resist *V. ceranae* invasion; an alternative may be that *V. ceranae* suppressed apoptosis by modulating these host-derived DEcircRNAs, the underlying mechanism of which is an attractive research direction.

### 4.4. A Working Model of DEcircRNA-Mediated Host Response to V. ceranae Infection

Based on the findings in this work, a model of the DEcircRNA-mediated response of *A. m. ligustica* workers’ midguts to *V. ceranae* infection is summarized in Figure 10. When *V. ceranae* infects the *A. m. ligustica* worker, partial DEcircRNAs in the host’s midgut epithelial cells are likely to regulate the transcription of the corresponding source genes, and some other DEcircRNAs may act as “miRNA sponges”, further participating in the regulation of host cell renewal and structure, material and energy metabolism, and cellular and humoral immune responses.

## 5. Conclusions

In this current work, a differential expression profile of circRNAs and the potential regulatory role of DEcircRNAs in *A. m. ligustica* workers’ midguts responding to *V. ceranae* infection, especially the hosts’ DEcircRNA-mediated immune defense, were comprehensively analyzed for the first time. Our data revealed that the overall expression of host circRNAs was altered due to *V. ceranae* infection, a portion of DEcircRNAs may participate in the response to *V. ceranae* by the host via regulation of the transcription of source genes, and some other DEcircRNAs were likely to regulate the host response as “molecular sponges” of miRNAs and further control the expression of downstream genes. Additionally, a complex regulatory network was formed by DEcircRNAs, DEmiRNAs, and DEmRNAs, which were jointly involved in host responses, including cellular and humoral immune responses. These findings provided not only a foundation for clarifying the molecular mechanism underlying the response of *A. m. ligustica* workers to *V. ceranae* invasion, but also a new insight into further understanding host–pathogen interaction during nosemosis.

## Figures and Tables

**Figure 1 biology-11-01285-f001:**
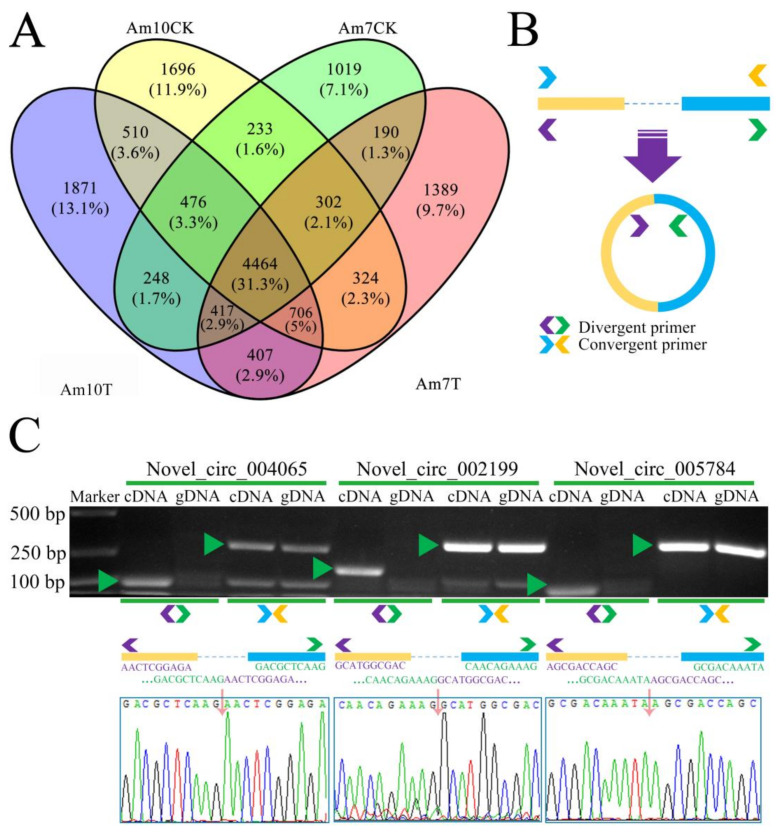
Venn analysis and molecular validation of *A. m. ligustica* circRNAs. (**A**) Venn analysis of circRNAs identified in four groups. Purple, yellow, green, and pink colors respectively represent circRNAs identified in Am10T, Am10CK, Am7CK, and Am7T groups. (**B**) A schematic diagram showing the designing of convergent primers and divergent primers for circRNAs. (**C**) AGE and Sanger sequence for amplified products from novel_circ_004065, novel_circ_002199, and novel_circ_005784 with divergent primers. Blue and orange arrows indicate convergent primers, while purple and green arrows indicate divergent primers. Green triangle within gel indicates target band. Pink arrowhead indicates back-splicing site. Yellow and blue rectangles represent flanking sequence around back-splicing site.

**Figure 2 biology-11-01285-f002:**
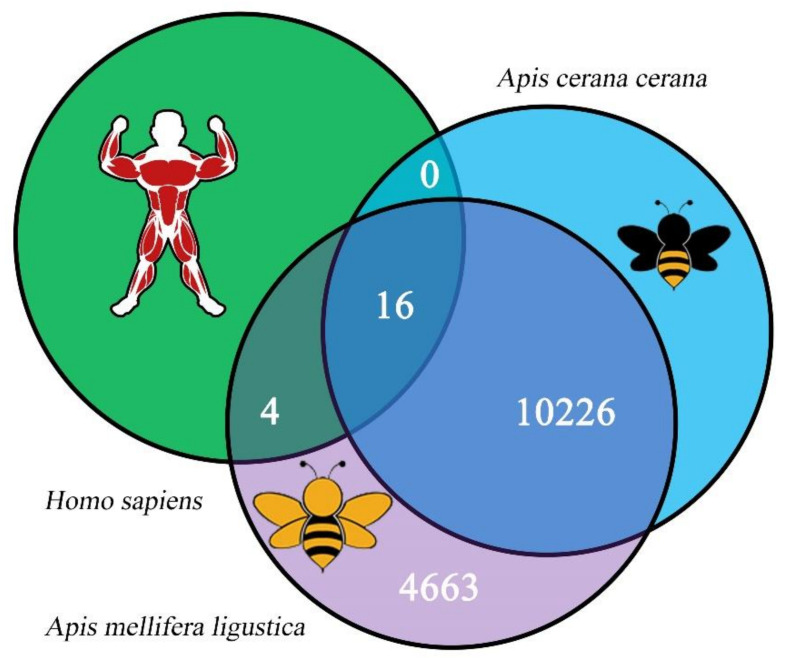
Sequence conservation of 14,909 *A. m. ligustica* circRNAs. Green circle indicates *H. sapiens* circRNAs, blue circle indicates *A. c. cerana* circRNAs, while purple circle indicates *A. m. ligustica* circRNAs.

**Figure 3 biology-11-01285-f003:**
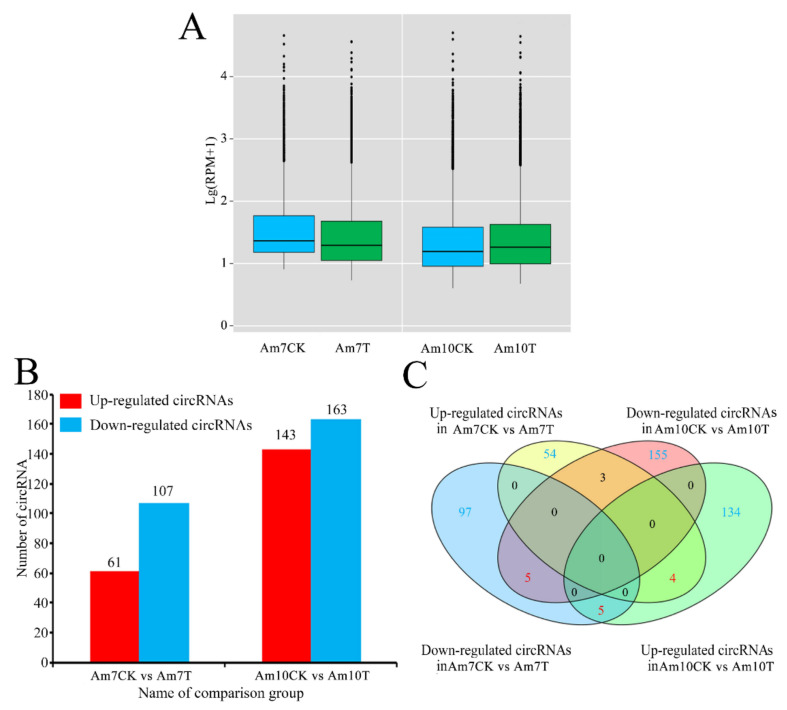
Differential expression profile of circRNAs in *V. ceranae*-inoculated and uninoculated midguts of *A. m. ligustica* workers. (**A**) Boxplots showing the overall expression levels of circRNAs identified in four groups. Blue boxplot indicates Am7CK or Am10CK group, while green boxplot indicates Am7CK or Am10CK group. (**B**) Number of DEcircRNAs in two comparison groups. Red and blue columns respectively indicate up-regulated and down-regulated circRNAs in the Am7CK vs. Am7T comparison group and Am10CK vs. Am10T comparison group). (**C**) Venn analysis of up-regulated and down-regulated circRNAs in the Am7CK vs. Am7T comparison group and in the Am10CK vs. Am10T comparison group (Blue and yellow circles indicate down-regulated and up-regulated circRNAs in Am7CK vs Am7T comparison group, while pink and green circles indicate down-regulated and up-regulated circRNAs in Am10CK vs Am10T comparison group).

**Figure 4 biology-11-01285-f004:**
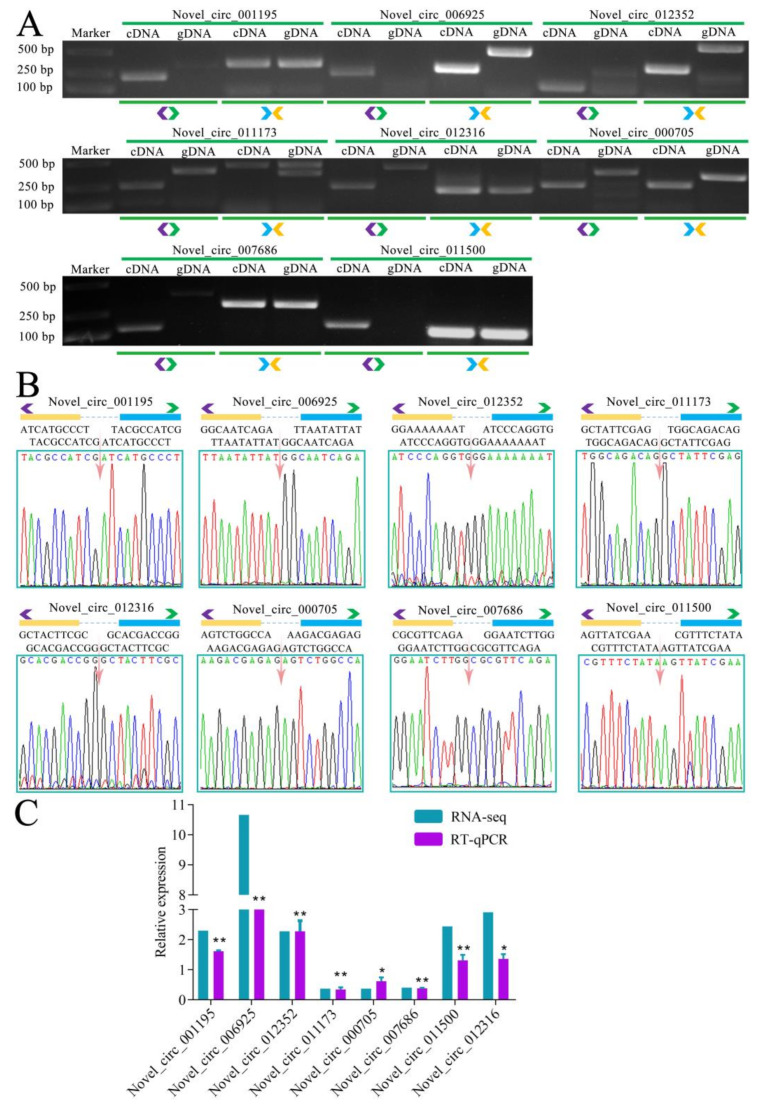
PCR, Sanger sequencing, and RT-qPCR validation of DEcircRNAs. (**A**) Agarose gel electrophoresis of products from PCR amplification of eight DEcircRNAs with specific divergent primers. Blue and orange triangle arrows indicate convergent primers, purple and green triangle arrows indicate divergent primers. (**B**) Sanger sequencing of amplified fragments from eight DEcircRNAs. Blue and orange triangle arrows indicate convergent primers, purple and green triangle arrows indicate divergent primers. pink arrowhead indicates back-splicing site. Yellow and blue rectangles represent flanking sequence around back-splicing site. (**C**) RT-qPCR results of eight DEcircRNAs (“*” indicates *p* ≤ 0.05, “**” indicates *p* ≤ 0.01).

**Figure 5 biology-11-01285-f005:**
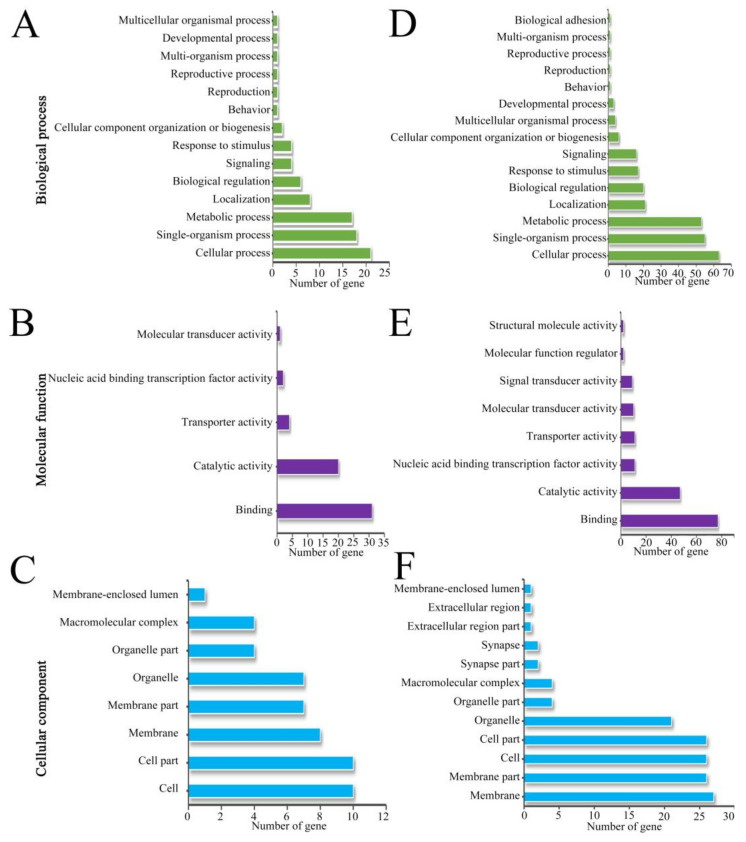
Functional annotation of DEcircRNAs’ source genes in the Am7CK vs. Am7T and Am10CK vs. Am10T comparison groups. (**A**–**C**) Biological-process-, molecular-function-, and cellular-component-associated terms annotated using DEcircRNAs’ source genes in the Am7CK vs. Am7T comparison group; (**D**–**F**) biological-process-, molecular-function-, and cellular-component-related terms annotated using DEcircRNAs’ source genes in the Am10CK vs. Am10T comparison group.

**Figure 6 biology-11-01285-f006:**
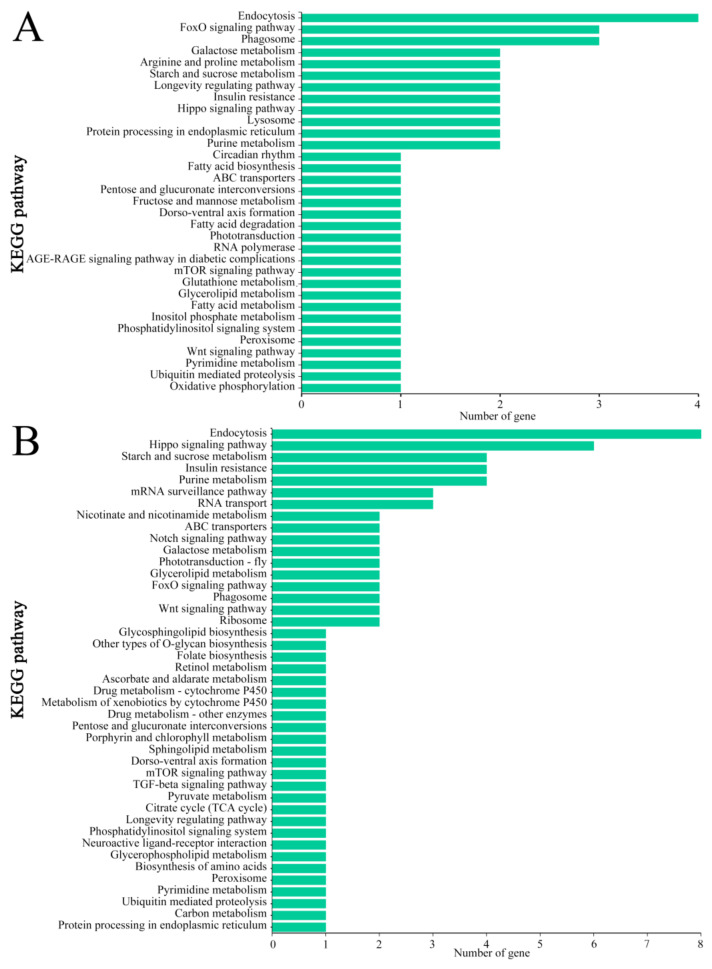
KEGG pathway annotation of source genes of DEcircRNAs in the Am7CK vs. Am7T and Am10CK vs. Am10T comparison groups. (**A**) Pathways annotated by DEcircRNAs’ source genes in the Am7CK vs. Am7T comparison group; (**B**) pathways annotated by DEcircRNAs’ source genes in the Am10CK vs. Am10T comparison group.

**Figure 7 biology-11-01285-f007:**
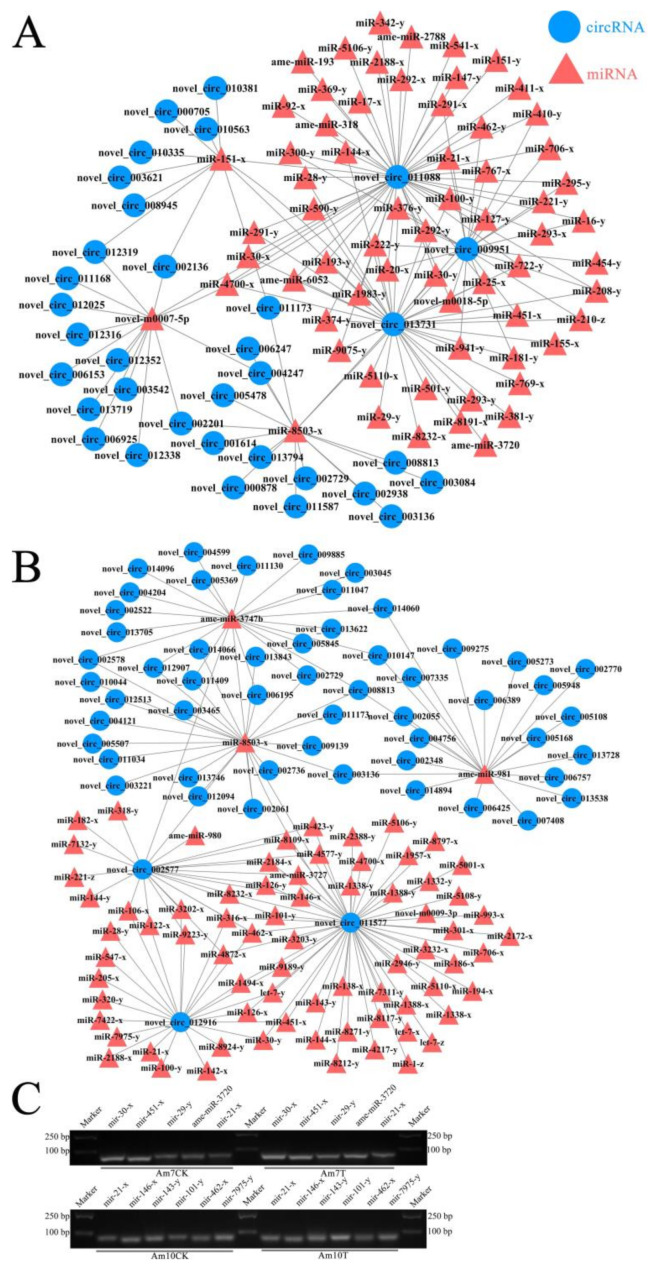
DEcircRNA-miRNA regulatory network. (**A**) Regulatory networks of DEcircRNAs and their target -miRNAs in the Am7CK vs. Am7T group; (**B**) regulatory networks of DEcircRNAs and their target -miRNAs in the Am10CK vs. Am10T group; (**C**) stem-loop PCR validation of DEcircRNA target miRNAs.

**Figure 8 biology-11-01285-f008:**
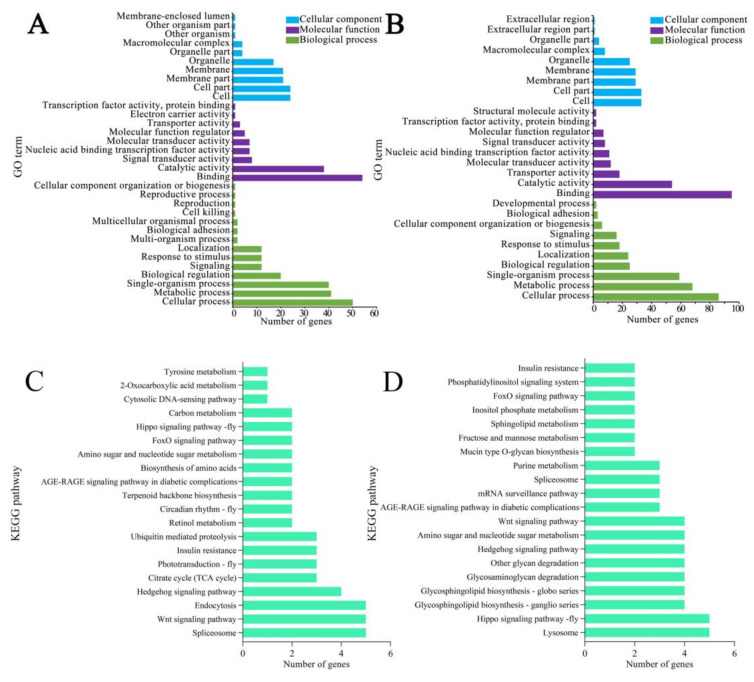
Function and pathway annotations of target mRNAs within ceRNA regulatory networks. (**A**,**B**) GO terms annotated by target mRNAs of DEcircRNA-targeted miRNAs in the Am7CK vs. Am7T comparison group. (**C**,**D**) top 20 KEGG pathways annotated by target mRNAs of DEcircRNA-targeted miRNAs in the Am10CK vs. Am10T comparison group.

**Figure 9 biology-11-01285-f009:**
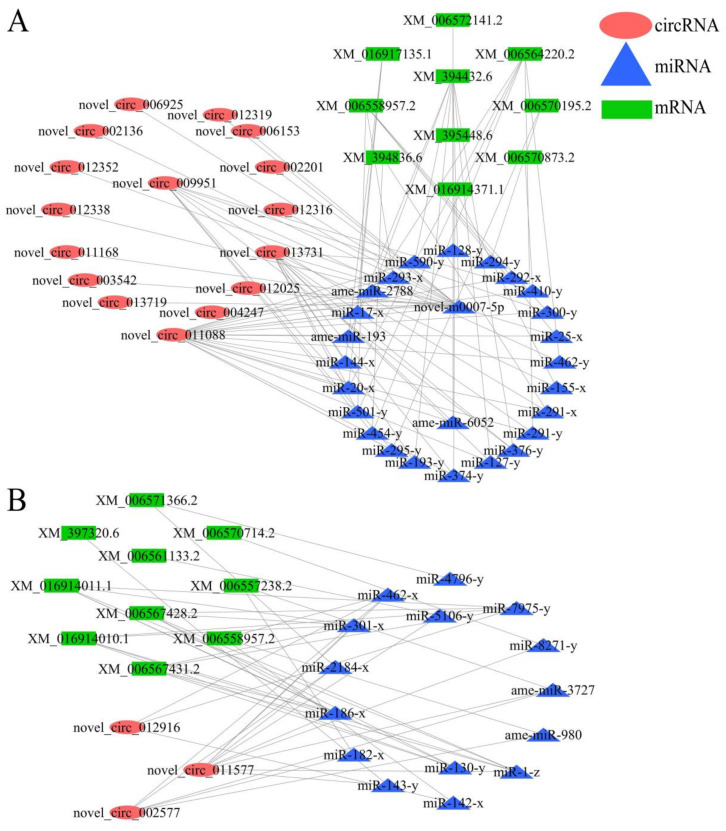
CeRNA regulatory networks relative to host cellular and humoral immunity. (**A**) Cellular and humoral immune-associated ceRNA network of DEcircRNAs in the Am7CK vs. Am7T group. (**B**) cellular and humoral immune-associated ceRNA network of DEcircRNAs in the Am10CK vs. Am10T group.

**Figure 10 biology-11-01285-f010:**
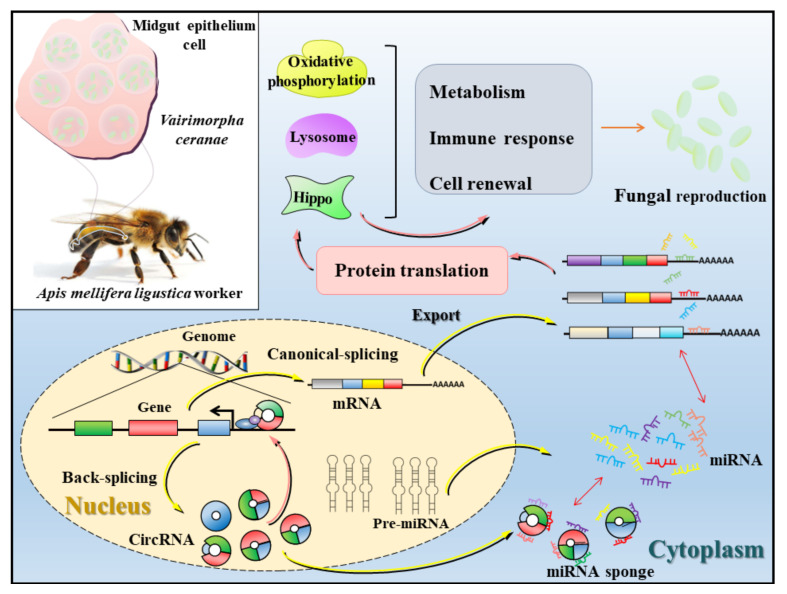
A working model showing DEcircRNA-mediated stress response of an *A. m. ligustica* worker’s midgut to *V. ceranae* infection. Green, red, blue, and grey rectangles in both nucleus and cytoplasm indicate different exons, while black lines indicate introns. Yellow, purple, and green irregular shapes represent three host pathways associated with energy metabolism and immune.

**Table 1 biology-11-01285-t001:** Mapping of anchor reads to the reference genome of *A*. *mellifera*.

Samples	Anchor Reads	Mapped Anchor Reads	Data Source
Am7T1	220,977,612	19,786,662 (8.95%)	This study
Am7T2	302,675,036	26,173,216 (8.65%)	This study
Am7T3	184,550,634	16,349,641 (8.86%)	This study
Am10T1	389,439,122	26,881,797 (6.90%)	This study
Am10T2	327,093,752	22,105,363 (6.76%)	This study
Am10T3	264,237,670	18,445,112 (6.98%)	This study
Am7CK1	236,400,908	19,006,601 (8.04%)	[40]
Am7CK2	168,420,472	19,873,468 (11.80%)	[40]
Am7CK3	136,779,122	17,502,228 (12.80%)	[40]
Am10CK1	195,020,440	22,441,139 (11.51%)	[40]
Am10CK2	184,091,234	21,109,367 (11.47%)	[40]
Am10CK3	168,790,774	17,765,330 (10.53%)	[40]

**Table 2 biology-11-01285-t002:** Source genes of DEcircRNAs engaged in host cellular and humoral immune pathways.

Immune Pathway	Number of Source Genes in Am7CK vs. Am7T	Number of Source Genes in Am10CK vs. Am10T	Ko Number
Endocytosis	4	8	ko04144
Phagosome	3	2	ko04145
Lysosome	2	0	ko04142
Ubiquitin-mediated proteolysis	0	1	ko04120
Metabolism of xenobiotics by cytochrome P450	0	1	ko00980
FoxO signaling pathway	3	2	ko04068

**Table 3 biology-11-01285-t003:** Summary of DEcircRNAs within ceRNA regulatory networks involved in host cellular and humoral immunity.

Pathway ID	Number of DEcircRNAs in Am7CK vs. Am7T	DEcircRNA ID	Number of DEcircRNAs in Am10CK vs. Am10T	DEcircRNA ID
Cellular immune-related pathway				
Endocytosis	3	novel_circ_011088, novel_circ_013731, novel_circ_009951	1	novel_circ_011577
Phagosome	2	novel_circ_011088, novel_circ_009951	0	
Ubiquitin-mediated proteolysis	3	novel_circ_011088, novel_circ_013731, novel_circ_009951	1	novel_circ_011577
Lysosome	0		2	novel_circ_011577, novel_circ_012916
Humoral immune-related pathway				
FoxO signaling pathway	16	novel_circ_011088, novel_circ_013731, novel_circ_009951, novel_circ_013719, novel_circ_012352, novel_circ_012338, novel_circ_012319, novel_circ_012316, novel_circ_012025, novel_circ_011168, novel_circ_006925, novel_circ_006153, novel_circ_004247, novel_circ_003542, novel_circ_002201, novel_circ_002136	3	novel_circ_011577, novel_circ_002577, novel_circ_012916
MAPK signaling pathway	2	novel_circ_013731, novel_circ_011088	2	novel_circ_011577, novel_circ_002577

## Data Availability

Raw reads were uploaded to the Short Read Archive (SAR) database of the NCBI under BioProject number PRJNA406998.

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
