# Peer review of "Deciphering the CircRNA-Regulated Response of Western Honey Bee (Apis mellifera) Workers to Microsporidian Invasion"

_biology, 2022, doi:10.3390/biology11091285_

Round 1

Reviewer 1 Report

The manuscript of the title “Deciphering the circRNA-regulated response of western honeybee workers to microsporidian invasion” provides the novel mechanism of circRNA-mediated immune responses of western honeybee workers that showed insight into pathogen and host interaction. Those molecules of circRNA discovered in this work can be directly used to control the problems from microsporidia to the western honeybee.  The introduction section is well-written with clear background and objectives. The section on materials and methods is well described with adequate information to follow. Results have clear illustrations and descriptions. The discussion part provides comprehensive information.

However, there are a few points that need to improve in the manuscript as follows.

Line 2 Please also include the scientific name to provide a unique name of the organism in the title.

Line 11-24 Please rewrite “Simple Summary” section. The current version just looks like it was copied and pasted from the abstract.

Line 12 N. ceranae is the first time presented in the article please provide the full name of the generic name.

Line 108 A full name of lncRNAs.

Line 189 Please remove the citation of (Kim et al., 2013) because it is provided with a numeric reference of 50.

Line 199 A full name of RPM.

Author Response

Reviewer 1

Line 2 Please also include the scientific name to provide a unique name of the organism in the title.

Response: A unique name of the organism in the title was provided.

1.Line 11-24 Please rewrite “Simple Summary” section. The current version just looks like it was copied and pasted from the abstract.

Response: Following your helpful suggestion, we rewrote the “Simple Summary” section, which was more concise and clear.

Line 12 N. ceranae is the first time presented in the article please provide the full name of the generic name.

Response: Correction was made according to your kind comment.

Line 108 A full name of lncRNAs.

Response: The full name of long noncoding RNAs (lncRNAs) was added here. Thanks.

Line 189 Please remove the citation of (Kim et al., 2013) because it is provided with a numeric reference of 50.

Response: The citation of (Kim et al., 2013) was deleted following your kind comment.

Line 199 A full name of RPM.

Response: We added the full name of RPM (reads per million) here in the revised manuscript.

Reviewer 2 Report

The work is devoted to an important problem - the study of nosematosis of honey bees. For the first time, the mechanism underlying the circRNA-mediated immune responses of western honey bee workers to N. ceranae infestation has been studied, as well as providing new insight into host-microsporidia interactions during nosemosis. The work expands the understanding of insect immunity.

The paper obtained important data on the immunology of honey bees, on the mechanisms that protect bees from pathogens. The results can be used to develop methods and mechanisms for protecting bee families from pathogens of various nature.

As a recommendation, I would like to suggest that the authors provide the names of the species with  Latin names, in accordance with the requirements of the Journal.

My congratulations to the authors. Thank you for the interesting article.  

Author Response

Reviewer 2

As a recommendation, I would like to suggest that the authors provide the names of the species with Latin names, in accordance with the requirements of the Journal.

Response: Thanks so much for your kind comment and helpful advice. We carefully checked the whole manuscript and provided the Latin names of the species.

Reviewer 3 Report

Non-coding RNAs, including circular RNAs, have been found recently to serve important role in cellular functions at molecular level related to gene expression. This knowledge opens new horizons in fundamental and applied molecular biology. In microsporidia-host interactions, these studies are scarce and need to be emphasized to ensure better understanding of biological processes and improvement of control of these notorious pathogens. The presented paper suits the journal scope and is of interest for the readers. The research is sound but several aspects need improvement.

1) Allocation of the microsporidium under study to the genus of Nosema is outdated as it doesn’t correspond to the molecular phylogeny and recently revised nomenclature (see https://pubmed.ncbi.nlm.nih.gov/31738888/ ).

2) In the paper under review, the authors imply that “this is the first documentation of circRNA-mediated immune response of honey bee to N. ceranae infection” (Lines 116-117). This seems to be true, as another publication (https://link.springer.com/article/10.1007/s00253-019-10159-9) describes the role of DEcircRNAs in immune response to invasion in general terms, not pertaining to a particular disease, though indicating importance for microsporidiosis and chalkbrood. And though the authors cite this paper in relation to methodology, the previous findings are not compared to the novel ones in terms of immune function regulation. Moreover, I cannot understand what is the reason of submission of two distinct preprints to bioarxiv.org with partially overlapping sets of authors: “CircRNA-regulated immune response of Asian honey bee workers to microsporidian infection” (https://www.biorxiv.org/content/10.1101/2022.06.30.498258v1.full.pdf) and “Deciphering the mechanism underlying 2 circRNA-mediated immune responses of western 3 honeybees to Nosema ceranae infection” (https://www.biorxiv.org/content/10.1101/2020.10.25.353938v1.full.pdf). Both of these “papers” pretend to be the first study of involvement of circRNAs in honey bee immune response to the microsporidiosis.

3) It is not obvious why, though indicating that cicrRNAs are conserved among a broad spectrum of animals, including various insects, authors study conservation only in two honey bee species and human

4) the format of biobliographical references should be double checked

5) Style and grammar need reconsideration, below are some examples

Line 88: circRNAs can resist the digestion of RNase R – “digestion of” or “digestion by”?

Lines 93-94: avoid word repetition (previous/previously)

Lines 110-116: break down the whole sentence for clarity

Line 113: were identified their source genes were – please rewrite for clarity

Line 115: “immune were” – please specify

Line 129: it’s = it was

Lines 131-132: please use “midguts of untreated workers”, not untreated midguts” which is senseless in this context

Line 177: past perfect is not justified here

Line 245-246: “mixture … was used as templates” (singular vs plural)

Line 501: “metabolisms” = “metabolism”

Author Response

Reviewer 3

1.Allocation of the microsporidium under study to the genus of Nosema is outdated as it doesn’t correspond to the molecular phylogeny and recently revised nomenclature (see https://pubmed.ncbi.nlm.nih.gov/31738888/ ).

Response: Thank You for your valuable comment. According to the recently revised nomenclature (Tokarev, Y.S.; Huang, W.F.; Solter, L.F.; Malysh, J.M.; Becnel, J.J.; Vossbrinck, C.R.; A formal redefinition of the genera Nosema and Vairimorpha (Microsporidia: Nosematidae) and reassignment of species based on molecular phylogenetics. J. Invertebr. Pathol. 2020, 169: 107279. doi: 10.1016/j.jip.2019.107279. Epub 2019 Nov 15. PMID: 31738888.), the genus of Nosema ceranae was Vairimorpha. Accordingly, Nosema ceranae was renamed Vairimorpha ceranae throughout the whole manuscript.

2.In the paper under review, the authors imply that “this is the first documentation of circRNA-mediated immune response of honey bee to N. ceranae infection” (Lines 116-117). This seems to be true, as another publication (https://link.springer.com/article/10.1007/s00253-019-10159-9) describes the role of DEcircRNAs in immune response to invasion in general terms, not pertaining to a particular disease, though indicating importance for microsporidiosis and chalkbrood. And though the authors cite this paper in relation to methodology, the previous findings are not compared to the novel ones in terms of immune function regulation. Moreover, I cannot understand what is the reason of submission of two distinct preprints to bioarxiv.org with partially overlapping sets of authors: “CircRNA-regulated immune response of Asian honey bee workers to microsporidian infection” (https://www.biorxiv.org/content/10.1101/2022.06.30.498258v1.full.pdf) and “Deciphering the mechanism underlying circRNA-mediated immune responses of western honeybees to Vairimorpha ceranae infection” (https://www.biorxiv.org/content/10.1101/2020.10.25.353938v1.full.pdf). Both of these “papers” pretend to be the first study of involvement of circRNAs in honey bee immune response to the microsporidiosis.

Response: Thank you so much for your valuable comment. In fact, these two preprints you mentioned were posted by our group, which focused on interaction between two bee species including western honey bee (Apis mellifera) and eastern honey bee (Apis cerana) and two fungal pathogen/parasite such as Vairimorpha ceranae (parasite of bee nosimosis) and Ascosphaera apis (pathogen of chalkbrood disease). Previously, we conducted characterization and investigation of circRNAs in A. mellifera and A. cerana, two sister bee species widely used in apicultural practice in China and many other countries, aiming to unclose immune responses of the aforementioned two bee species and compare their difference, so we posted two preprints in bioRxiv.

On basis of the date of publication, this study should be the first report of involvement of circRNAs in honey bee immune response to N. ceranae infection. Accordingly, we will modify related description in “CircRNA-regulated immune response of Asian honey bee workers to microsporidian infection”. In view of that this manuscript was posted ahead of the preprint “CircRNA-regulated immune response of Asian honey bee workers to microsporidian infection”, we compared the findings in this current work with the results in our another published article associated with A. cerana circRNAs (https://link.springer.com/article/10.1007/s00253-019-10159-9) in terms of A. cerana immune defense. It’s observed that in midgut of A. cerana, target genes of circRNAs were annotated to immune system process and nine immune-related pathways (FoxO signaling pathway, Phagosome, Lysosome, Endocytosis, Drug metabolism-cytochrome P450, Metabolism of xenobiotics by cytochrome P450, MAPK signaling pathway–fly, Jak-STAT signaling pathway, and Ubiquitin mediated proteolysis). Comparatively, target genes of DEcircRNAs in this study were annotated to response to stimulus and six immune-related pathways (Endocytosis, Phagosome, Lysosome, Ubiquitin mediated proteolysis, Metabolism of xenobiotics by cytochrome P450, and FoxO signaling pathway). The comparison indicated that circRNAs in A. mellifera and A. cerana were potentially involved in the immune defense system and six immune-related pathways (Endocytosis, Phagosome, Lysosome, Ubiquitin mediated proteolysis, Metabolism of xenobiotics by cytochrome P450, and FoxO signaling pathway) and further participate in host immune response to N. ceranae invasion.

Additionally, contents of comparison of circRNAs and circRNA-regulated immune responses between A. cerana and A. mellifera were added in the revised manuscript.

3.It is not obvious why, though indicating that circRNAs are conserved among a broad spectrum of animals, including various insects, authors study conservation only in two honey bee species and human

Response: Currently, study on insect circRNAs is limited. Since we were interested in conservation among circRNAs in various insects, when performing conservation analysis of circRNAs, we tried to download sequences of reported cicrRNAs’ source genes in other insects including Drosophila melanogaster, Bombyx mori, and honey bee, but we found only raw data from deep sequencing was deposited in public databases such as NCBI SRA, sequences of circRNAs’ source genes were unavailable. Therefore, we conducted conservation investigation of circRNAs in A. mellifera, A. cerana, and human, the result could reveal conservation among circRNAs in insects and mammals to some extent. Accordingly, we made necessary modification in the revised version of manuscript and we believe that the conclusion “16 circRNAs (novel_circ_002387, novel_circ_008844, novel_circ_008846, nov-el_circ_008847, etc.) were highly conserved in all three species mentioned above, suggestive of pivotal function of them in both honey bees and human, thus deserving additional investigation”

4.the format of biobliographical references should be double checked

Response: Following your comment of importance, the format of each reference was seriously examined followed by necessary correction.

5.Style and grammar need reconsideration, below are some examples

Line 88: circRNAs can resist the digestion of RNase R – “digestion of” or “digestion by”?

Response: “RNase R” you suggested is more suitable, so we made modification here. Thanks.

Lines 93-94: avoid word repetition (previous/previously)

Response: We carefully checked the whole manuscript and corrected all word repetition.

Lines 110-116: break down the whole sentence for clarity

Response: The sentence was broken down following your kind recommendation.

Line 113: were identified their source genes were – please rewrite for clarity

Response: This sentence was modified todifferentially expressed circRNAs (DE-circRNAs) were identified and their source genes were then annotated”.

Line 115: “immune were” – please specify

Response: According to your kind comment, it was replaced byand circRNAs, associated with host cellular and humoral immune, were further explored”.

Line 129: it’s = it was

Response: This is corrected in the revised manuscript.

Lines 131-132: please use “midguts of untreated workers”, not untreated midguts” which is senseless in this context

Response: Correction was made following your kind comment in the revised manuscript.

Line 177: past perfect is not justified here

Response: It’s a mistake here, which was corrected in the revised manuscript.

Line 245-246: “mixture … was used as templates” (singular vs plural)

Response: The description was modified to “mixture … was used as template”. Thanks.

Line 501: “metabolisms” = “metabolism”

Response: “metabolisms” was replaced by “metabolism” based on your kind comment.

Reviewer 4 Report

In this manuscript, the authors primarily identified and detailed transcriptional patterns of circular RNAs (circRNA) in the gut of Nosema-infected Apis mellifera ligustica; they also examined similarity of those circRNAs to human and Ap. cerana, and made various predictions regarding putative or potential interactions. The work advances what is known regarding circRNAs, particularly the identification of a new set of these RNAs from an important insect.

There may be a significant flaw, fatal to the broader considerations, specifically the communicated differential expression and related inferences. Namely here, the authors poorly communicate what de novo sequencing work they have performed in this project as compared to previously published data sets. It appears (Table 1) that the sequencing work here consists only of nosema infected tissue, while all control tissue came from study referenced in citation 40. If that is the case, the experiment appears ill-controlled in that biological and technical errors (due to carrying the work out at different times, seasons, hive conditions, etc, or with different batches of reagents, quality of machines in service cycles, etc) may result in biased outputs in the experimental (T-groups) vs control (CK-groups) data. So, overall, the authors fail to adequately clearly communicate what RNASeq samples were novel for this study and which were not; if this were clearly communicated, it might be that the comparisons (Nosema vs check) are legitimately controlled. So, the authors need to clarify this and if they are comparing separately performed data analyses (Check from previous isolations ref 40, Nosema from current) they ought to identify and discuss possible errors. 

Additionally, the authors’ approach to identifying circRNAs seems problematic to me. They cite previous approach (29, line 187) in that they mapped the reads to A mellifera. They then note that they performed unmapped read analyses via reference sequences (189-190) and unbiased search (191-194). Subsequently, though, they note that a significant number of reads didn’t map to cerana (30+%). Given this, there are no details on how many of the predicted circRNAs might be from Nosema. If they are from Nosema, what is the significance? 

Beyond the above issues, the paper had numerous more minor issues: 

-Throughout the authors use a convention of “number (number)”. While defined in Summary and Abstract it ought to be defined later, as well.

-What are ceRNAs? This is not defined at first (line 160) or subsequent use.

-Lines 61-63: the authors imply that CCD drives colony reproduction and survival, while I believe it is the other way around (CCD is a manifestation of numerous characteristics like reproductive rate, survival rate, absconding rate, etc).

-195-197: no details were performed on comparisons of A. mellifera circRNAs to H sapiens and A. cerana. Minimally, access dates and algorithm should be provided. 

-227-230: are these novel samples, or pulled from earlier samples? Should be clearer.  If former, then details need to be provided (number and nature of samples, number of replicates).

-239, 244: I thought the notation of “7 (10) dpi” was as above referring to number of samples at time point, but then realized it is not. Again, more clarity would be beneficial to the readability of the paper. Perhaps “The total RNA of worker’s midguts at 7 and 10 dpi was isolated….”

-257: how were miRNAs randomly selected? 

-264-266: as above (227)– were these novel or previously obtained samples? If former, what are the details?

-280: as above (227, 264) – nature of these isolates.

-293: were primer/reaction efficiencies validated?

-294: what does “three parallel replicates” mean? Is parallel here the same as technical or well replicate? 

-Section 3.2: the wording describing the number of circRNAs and their novel vs unique nature is ambiguous. Particularly, the authors state that 14,909 circRNAs were identified – these are nonredundant circRNAs pulled from the total of 29729 (= 8199 + 8711 + 6530 + 6289)?

-Fig 2: some sharing of sequences by H sapiens and A cerana is implied but no number is provided. 

-Figs 5-6: the descriptions for both of these figures are unclear. What does it mean that B represents “comparison groups”? I’m not sure what A refers to. Additionally, the text and title suggest that the functional groups are differentially expressed ,but there is no indication of how much (understandable due to the binning approach used) but importantly whether the DE means up or down relative to for example control. 

-lines 496-505: the model is attractive, however the text (496-502) is nothing new and seems like it should be in the discussion not results.

-line 549: unclear what “21 and two source genes” means

-lines 709-711: the authors conclude that DEcircRNAs may regulate apoptosis to induce an apoptotic response. An alternative may be that nosema induces these circRNAs leading to inhibition of apoptosis. What evidence suggests the authors’ rather than this alternative interpretation?

Author Response

Reviewer 4

1.There may be a significant flaw, fatal to the broader considerations, specifically the communicated differential expression and related inferences. Namely here, the authors poorly communicate what de novo sequencing work they have performed in this project as compared to previously published data sets. It appears (Table 1) that the sequencing work here consists only of nosema infected tissue, while all control tissue came from study referenced in citation 40. If that is the case, the experiment appears ill-controlled in that biological and technical errors (due to carrying the work out at different times, seasons, hive conditions, etc, or with different batches of reagents, quality of machines in service cycles, etc) may result in biased outputs in the experimental (T-groups) vs control (CK-groups) data. So, overall, the authors fail to adequately clearly communicate what RNASeq samples were novel for this study and which were not; if this were clearly communicated, it might be that the comparisons (Nosema vs check) are legitimately controlled. So, the authors need to clarify this and if they are comparing separately performed data analyses (Check from previous isolations ref 40, Nosema from current) they ought to identify and discuss possible errors.

Response: Thanks for your comments of great importance. Description regarding biological samples for deep sequencing were unclear in the original version of manuscript. In fact, both un-infected samples and N. ceranae-infected samples were previously prepared in the same experiment in our lab. Firstly, in view of the limited information about A. mellifera circRNAs, when the sequencing project was finished, we conduced identification, characterization, and validation of A. mellifera circRNAs based on sequencing data from un-infected workers’ midgut tissues, in which N.ceranae-derived data could be excluded (reference 40). Secondly, in this work, we performed investigation of A. mellifera workers’ responses to N. ceranae infection on basis of data from un-infected and N. ceranae-infected workers’ midgut tissues. Accordingly, we seriously improved corresponding description in the revised manuscript.

2.Additionally, the authors’ approach to identifying circRNAs seems problematic to me. They cite previous approach (29, line 187) in that they mapped the reads to A. mellifera. They then note that they performed unmapped read analyses via reference sequences (189-190) and unbiased search (191-194). Subsequently, though, they note that a significant number of reads didn’t map to A. mellifera (30+%). Given this, there are no details on how many of the predicted circRNAs might be from Nosema. If they are from Nosema, what is the significance?

Response: Thanks for your valuable comment, following which we used the similar circRNA identification protocol to performed identification of V. ceranae circRNAs based on sequencing data from Am7T and Am10T groups (midguts of V. ceranae -inoculated workers at 7 dpi and 10 dpi); subsequently, 20 bp left anchors reads of V. ceranae circRNAs were mapped to 20 bp left anchors reads of A. m. lugustica circRNAs, while 20 bp right anchors reads of V. ceranae circRNAs were mapped to 20 bp right anchors reads of A. m. lugustica circRNAs, the result showed a no circRNA was shared by A. m. lugustica and V. ceranae (Table S1).

  1. Throughout the authors use a convention of “number (number)”. While defined in Summary and Abstract, it ought to be defined later, as well.

Response: Following your helpful suggestion, the definition of “number (number)” was added in Result section and Discussion section.

4.What are ceRNAs? This is not defined at first (line 160) or subsequent use.

Response: The full name of ceRNA is “competing endogenous RNA”. It’s defined at the beginning of the manuscript according to your kind comment.

5.Lines 61-63: the authors imply that CCD drives colony reproduction and survival, while I believe it is the other way around (CCD is a manifestation of numerous characteristics like reproductive rate, survival rate, absconding rate, etc).

Response: Thank for your valuable comment, following which we modified related description to “Evidently, N. ceranae has proved to be relevant to colony collapse disorder (CCD) [10-11], which severely influences many aspects of the colony, including reproductive rate, survival rate, absconding rate, etc [12-13].”

6.195-197: no details were performed on comparisons of A. mellifera circRNAs to H sapiens and A. cerana. Minimally, access dates and algorithm should be provided.

Response: Necessary information about conservative comparison of circRNAs among A. mellifera, A. cerana, and H. sapiens were added in Method section in the revised manuscript.

7.227-230: are these novel samples, or pulled from earlier samples? Should be clearer. If former, then details need to be provided (number and nature of samples, number of replicates).

Response: These were pulled from earlier samples. Necessary information was added here to improve the clarity.

8.239, 244: I thought the notation of “7 (10) dpi” was as above referring to number of samples at time point, but then realized it is not. Again, more clarity would be beneficial to the readability of the paper. Perhaps “The total RNA of worker’s midguts at 7 and 10 dpi was isolated….”

Response: Following your significant comments, we seriously checked the manuscript and made modifications to improve the clarity. Please see the revised version of manuscript.

9.257: how were miRNAs randomly selected?

Response: Here, based on targeting relationship between DEcircRNAs and miRNAs, mir-30-x targeted by novel_circ_000626, mir-451-x targeted by novel_circ_001614, mir-29-y targeted by novel_circ_000626, ame-miR-3720 targeted by novel_circ_004821, and mir-21-x targeted by novel_circ_005286 in Am7CK vs Am7T comparison group; mir-21-x targeted by novel_circ_002578, mir-146-x targeted by novel_circ_002142, mir-143-y targeted by novel_circ_000469, mir-101-y targeted by novel_circ_001441, mir-462-x targeted by novel_circ_002577, and mir-7975-y targeted by novel_circ_002736 in Am10CK vs Am10T comparison group were randomly selected for Stem-loop RT-PCR validation. It’s believed that the validation result of random selection of target miRNAs would be more reliable. Thanks.

10.264-266: as above (227)– were these novel or previously obtained samples? If former, what are the details?

Response: These were previously obtained samples.

11.280: as above (227, 264) – nature of these isolates.

Response: These isolates were pulled from earlier samples.

12.293: were primer/reaction efficiencies validated?

Response: When performing RT-qPCR validation of DEcircRNAs, the reaction was efficient and stable on basis of the melting curve after each reaction was finished, and there was always only one amplification peak showed in the melting curve, indicative of the specificity of used primers.

13.294: what does “three parallel replicates” mean? Is parallel here the same as technical or well replicate?

Response: Thank you for your kind comment, “three parallel replicates” here means “three technical replicates”.

14.Section 3.2: the wording describing the number of circRNAs and their novel vs unique nature is ambiguous. Particularly, the authors state that 14,909 circRNAs were identified – these are nonredundant circRNAs pulled from the total of 29729 (= 8199 + 8711 + 6530 + 6289)?

Response: In fact, 8,199 and 8,711 circRNAs were identified in Am7T and Am10T groups, while 6,530 and 6,289 circRNAs were respectively identified in Am7CK and Am10CK groups. Since some circRNAs identified in one group may also appear in another groups, after removing redundant circRNAs, a total of 14,909 nonredundant circRNAs in A. mellifera were identified. Accordingly, we improved the description to give more clear information.

15.Fig 2: some sharing of sequences by H sapiens and A cerana is implied but no number is provided.

Response: Based on your kind comment, we added numbers of H. sapiens circRNAs and A. cerana circRNAs in this section.

16.Figs 5-6: the descriptions for both of these figures are unclear. What does it mean that B represents “comparison groups”? I’m not sure what A refers to. Additionally, the text and title suggest that the functional groups are differentially expressed, but there is no indication of how much (understandable due to the binning approach used) but importantly whether the DE means up or down relative to for example control.

Response: After serious examination, we found there were mistakes here. Titles and legends of Figure 5 and Figure 6 were modified following your helpful comment in the revised manuscript. Thanks.

17.lines 496-505: the model is attractive, however the text (496-502) is nothing new and seems like it should be in the discussion not results.

Response: Findings in this current work were summarized in Figure 10, we agreed with you that it’s more suitable to transfer the working model to the discussion section. Thanks.

18.line 549: unclear what “21 and two source genes” means

Response: To make this sentence more concise, we modified this sentence in the revised manuscript.

19.lines 709-711: the authors conclude that DEcircRNAs may regulate apoptosis to induce an apoptotic response. An alternative may be that nosema induces these circRNAs leading to inhibition of apoptosis. What evidence suggests the authors’ rather than this alternative interpretation?

Response: Thanks for your important comment. It’s reported that N. ceranae was able to inhibit apoptosis in honey bees to facilitate microsporidian proliferation. After checking related papers and serious communication among authors, it’s hard to make the conclusion that “these DEcircRNAs may participate in activating apoptosis of honey bee to resist N. ceranae invasion”. In the revised version of manuscript, we modified this conclusion and improved the content.

Round 2

Reviewer 4 Report

In this revision, the authors have - through modification of the manuscript and response to reviewer - sufficiently addressed my major concerns with this submission. I thus have no major concerns. 

Minor concern: the figures and captions for Figs 3, 5, 6, and 8 all address differential expression. However, the captions are unclear for all. For Fig 3, the authors do not state what is the reference and thus denominator, versus which treatment represents an up- or down-regulation (I assume it's T / CK, but not explicit). In Fig 5, what are comparison groups? I also assume that the direction (up- vs down-) is irrelevant but an alternative depiction would be to show the number of upregulated in T vs CK to the right and down regulated to the left for each function. Fig 6 - what's Comparison groups (B)? Same for Fig 8.

Author Response

Response: Thanks so much for your helpful comments. We seriously checked Figures 3, 5, 6 and 8 as well as associated captions, and as you said, some necessary information was missing, which made these figures unclear. Accordingly, we modified captions of these figures and added necessary information into the revised version of manuscript. It’s believed that information about Figures 3, 5, 6 and 8 is clearer. As for Am7CK vs Am7T and Am10CK vs Am10T comparison groups, treatment groups were put behind control groups, indicating the expression level of a circRNA was up-regulated (down-regulated) in treatment group compared with the expression level in control group.